# Large-scale design and refinement of stable proteins using sequence-only models

**Jedediah M. Singer**[1]*, **Scott Novotney**[1], **Devin Strickland**[2], **Hugh K. Haddox**[3], **Nicholas Leiby**[1], **Gabriel J. Rocklin**[4], **Cameron M. Chow**[3], **Anindya Roy**[3], **Asim K. Bera**[3], **Francis C. Motta**[5], **Longxing Cao**[3], **Eva-Maria Strauch**[6], **Tamuka M. Chidyausiku**[3], **Alex Ford**[3], **Ethan Ho**[7], **Alexander Zaitzeff**[1], **Craig O. Mackenzie**[8], **Hamed Eramian**[9], **Frank DiMaio**[3], **Gevorg Grigoryan**[10], **Matthew Vaughn**[7], **Lance J. Stewart**[3], **David Baker**[3], **Eric Klavins**[2]

**1** Two Six Technologies, Arlington, Virginia, United States of America, **2** Department of Electrical and Computer Engineering, University of Washington, Seattle, Washington, United States of America, **3** Department of Biochemistry and Institute for Protein Design, University of Washington, Seattle, Washington, United States of America, **4** Department of Pharmacology and Center for Synthetic Biology, Northwestern University Feinberg School of Medicine, Chicago, Illinois, United States of America, **5** Department of Mathematical Sciences, Florida Atlantic University, Boca Raton, Florida, United States of America, **6** Department of Pharmaceutical and Biomedical Sciences, University of Georgia, Athens, Georgia, United States of America, **7** Texas Advanced Computing Center, Austin, Texas, United States of America, **8** Quantitative Biomedical Sciences Graduate Program, Dartmouth College, Hanover, New Hampshire, United States of America, **9** Netrias, Cambridge, Massachusetts, United States of America, **10** Departments of Computer Science and Biological Sciences, Dartmouth College, Hanover, New Hampshire, United States of America

\* jed.singer@twosixtech.com

**Data Availability Statement:** Models, code, and data are available at https://zenodo.org/record/4906529. The DOI is 10.5281/zenodo.4906529.

## Abstract

Engineered proteins generally must possess a stable structure in order to achieve their designed function. Stable designs, however, are astronomically rare within the space of all possible amino acid sequences. As a consequence, many designs must be tested computationally and experimentally in order to find stable ones, which is expensive in terms of time and resources. Here we use a high-throughput, low-fidelity assay to experimentally evaluate the stability of approximately 200,000 novel proteins. These include a wide range of sequence perturbations, providing a baseline for future work in the field. We build a neural network model that predicts protein stability given only sequences of amino acids, and compare its performance to the assayed values. We also report another network model that is able to generate the amino acid sequences of novel stable proteins given requested secondary sequences. Finally, we show that the predictive model—despite weaknesses including a noisy data set—can be used to substantially increase the stability of both expert-designed and model-generated proteins.

## Introduction

Most proteins, natural or designed, require a stable tertiary structure for functions such as binding [1], catalysis [2], or self-assembly [3]. Because structural stability derives from thousands of interactions between atoms, both attractive and repulsive, whose net sum is close to

**Funding:** This material is based upon work supported by the Defense Advanced Research Projects Agency (DARPA) and the Air Force Research Laboratory under Contract No. FA8750-17-C-0231 (and related contracts by SD2 Publication Consortium Members). The specified contract number applies to JMS, SN, NL, AZ, and HE, while related contracts under the same program pertain to other authors. We thank the staff at Northeastern Collaborative Access Team (NECAT) at Advanced Photon Source for the beamtime. Representatives of DARPA—the funders—asked interested scientific questions that may have provided ideas for study design and data analysis, but played no other role in study design or data analysis. They had no role in data collection or preparation of the manuscript. They encouraged publication after the decision to publish was made by the authors.

**Competing interests:** JMS and AZ are employed by Two Six Technologies, which has filed a patent on a portion of the technology described in this manuscript. This does not alter our adherence to PLOS ONE policies on sharing data and materials.

zero, precise prediction of stability is extremely challenging. Current approaches use combinations of statistical and physics-based potentials to approximate and refine calculations of these interactions. While much progress has been made, these programs comprise many terms that scale unfavorably in compute time, are imperfectly parameterized, and attempt to model poorly understood processes such as the entropy of the unfolded state or the polarizability of some atoms [4, 5]. Thus, calculating the stability of even a small protein is both computationally expensive and usually insufficiently accurate for practical use. This means that creating a small number of successful proteins typically requires the design and evaluation of a large number of candidates, at significant cost. Data-driven approaches, especially neural networks, implicitly solve only the calculations necessary to arrive at a prediction, and have been used in computational prediction tasks in other domains lacking accurate statistical or physics-based models. A successful application to protein stability would lead to both higher design-build-test throughput and higher accuracy in design for a wide range of applications.

There has been a recent surge in the application of machine learning to understanding properties of proteins. In particular, there is substantial interest in predicting the forces guiding protein folding and dynamics, leading to physically plausible models of protein structure [6]. [7] presents an approach for learning protein structure from primary sequence leveraging geometric constraints, which yields competitive accuracy with high computational performance. [8, 9] applied similar geometric constraints to a model that augmented its primary sequence input with data about mutation correlations in homologous proteins to yield state-of-the-art performance in structure prediction. There have been a few attempts to use primarily or exclusively primary sequence data to make predictions about proteins, leaving the underlying biology, chemistry, and physics to be learned implicitly. [10] showed that a greedy search guided by simple sequence-based machine-learning models could substantially improve the efficiency of guided evolution for protein binding. [11] developed a sequence-based recurrent model which generated fixed-length vector embeddings of proteins, based solely on primary sequence. They demonstrated that these embeddings were useful for solving complex problems in protein engineering, including stability prediction, using very simple models. [12–14] explored attention-based models that had been trained on large sets of protein sequences and found that the models' learned representations aligned with such biophysical properties as 3D structure and binding domains. In [15] an ensemble of simple machine learning models predicted stability based on features of protein secondary and tertiary structure, after training on a limited set of natural proteins. [16] used a regression model over Fourier transforms of encoded biophysical properties of primary sequences to identify mutations of a particular enzyme that increased stability while decreasing detrimental aggregation.

Data-driven models have also recently been applied to the direct design of proteins. [17] used recurrent neural networks to generate novel peptide designs based on a training set of helical antimicrobial peptides, with results that were predicted to be more active than random draws from the training set distribution. These designs were not tested in the laboratory, however. In [18], constraints derived from evolutionary statistics of single and paired amino acids yielded designs that demonstrated the intended enzymatic activity. [19] demonstrated that a model trained to predict 3D structure could be used to "hallucinate" novel sequences that formed experimentally validated proteins.

One drawback of machine learning models is that they require large amounts of training data. This presents a particular problem for modeling protein stability because, historically, experimental measurements of stability have been laborious [20–22]. Many previous applications of machine learning to problems in protein science have taken advantage of existing large datasets such as the Protein Data Bank [23], UniProt [24], or a handful of existing datasets of empirical measurements (e.g. [25, 26]). Such efforts can only address questions for

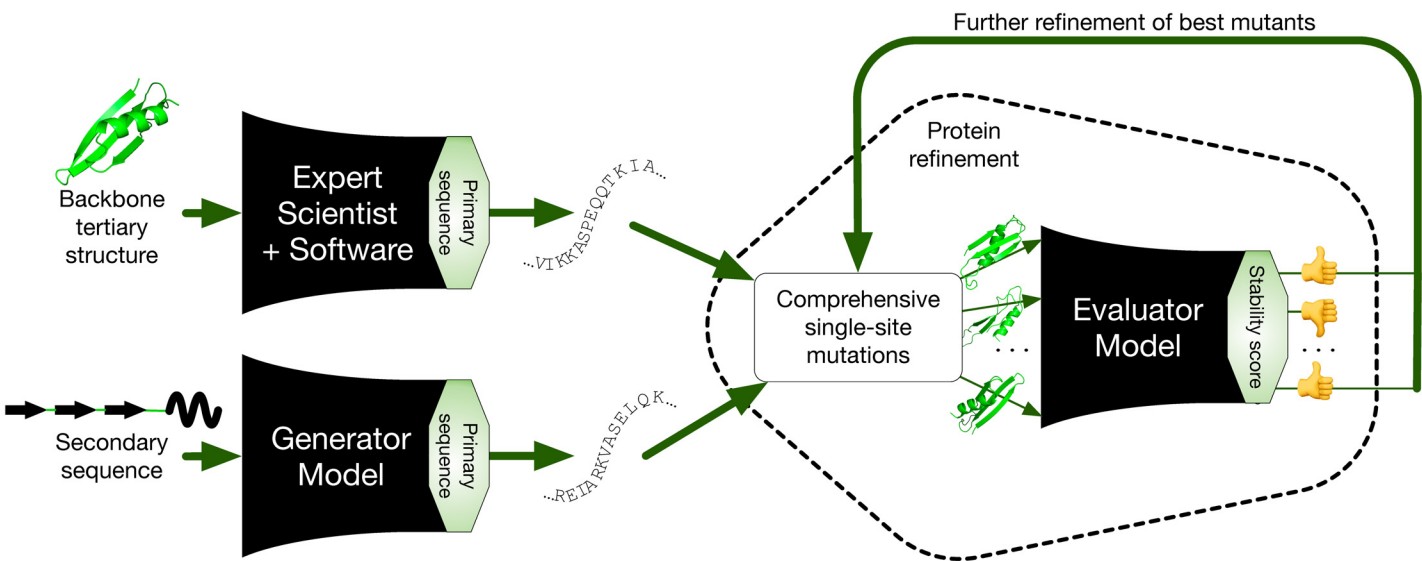

**Fig 1. Overview of design and refinement.** Proteins are designed, either by an expert using Rosetta or dTERMen software or by a neural network model that transforms secondary sequences into primary sequences. These designs are refined to maximize stability via an iterative procedure. At each step, the stability of all possible single-site substitutions is predicted by another neural network model. The mutants with the highest predicted stability are saved and used as seeds for the next round of optimization.

which there are pertinent data, and typically cannot test model predictions comprehensively on new data or fully support iterative or reinforcement learning strategies. Machine learning in combination with the generation of new data has yielded new insights, for example in the prediction of functional motifs such as alternative polyadenylation [27], alternative splicing [28], small molecule inhibitor discovery [29], and protein-protein interaction surfaces [30].

Here we describe a data-driven process for the high-speed, automated creation of stable protein designs (Fig 1). We use a previously described combination [26] of parallel oligo library synthesis, yeast surface display, and next-generation sequencing, coupled with Aquarium lab automation software (aquarium.bio) to generate very large datasets suitable for training and testing a neural network model of protein stability. We describe a computationally efficient neural network, the Evaluator Model (EM), that is able to predict protein stability with high accuracy, based solely on amino acid sequence. We validate the EM using data on almost 200,000 new protein designs, assayed using this robust experimental pipeline. We also demonstrate the use of the EM to refine the stability of protein designs by making multiple changes, increasing stability tenfold as evaluated by the assay described in [26]. We show that these refinements can easily be made to respect additional constraints on how they change the proteins, which in conjunction with other tools could lead to preservation of structure or function. We describe another neural network, the Generator Model (GM), that is able to create novel stable protein sequences at high speed; these sequences can also be successfully refined by the EM. Finally, we demonstrate via low-resolution methods that selected examples fold into stable structures, and report a high resolution crystal structure of one design that matches the expected topology.

## Results

### High-throughput measurement of protein stability

In order to generate enough data to train a sequence-based neural network model of stability, we adapted and automated a recently developed technique for assaying the stability of small

proteins through their relative resistance to digestion by proteases (S1 Fig). This approach uses DNA oligonucleotide gene library synthesis to encode the designed proteins, which are expressed in a yeast surface display system so that each cell displays many copies of a single designed protein. The yeast library is treated with trypsin and chymotrypsin in a range of concentrations, and sorted in a fluorescence activated cell sorter (FACS) to collect yeast cells with a high proportion of undigested protein. The resulting pools are then deep-sequenced, and the relative resistance to protease digestion is computed from the frequencies of sequence reads. We used this combination of the existing assay and software-based automation to evaluate the stability of up to 100,000 proteins in a single experiment.

While resistance to digestion by protease is partly determined by a protein's unfolding free energy (i.e. its stability), this relationship is complicated by the intrinsic, unfolded-state resistance of the sequence to cleavage. [26] devised a method to subtract the intrinsic resistance predicted by an unfolded-state model (USM) from the overall observed resistance. This yields a stability score that, in principle, represents only the component of resistance derived from folding. Because this correction is imperfect—for example, it only modestly improves agreement between trypsin- and chymotrypsin-derived stability scores—we reasoned that potential latent factors, such as the propensity of the sequence to form residual unfolded-state structures incompatible with binding to the active site of the protease, could also affect the cleavage resistance. We developed a more flexible USM to account for these additional factors. This new USM yielded predictions that improved upon the original USM in several metrics (S2 Fig). We also confirmed that the stability score calculated using the new USM was an improvement over raw $EC_{50}$, by comparing relationships between $EC_{50}$ values for the two proteases (S3 Fig) and relationships between stability scores for the two proteases (S4 Fig). Given the apparent improvements in stability score over raw $EC_{50}$, and in the new USM over the original, we chose to use a stability score based on the new USM for all analyses. Analyses performed with the original USM yield similar results and unchanged conclusions.

## Predicting stability with a sequence-only model

We built a convolutional neural network (CNN) model, which we call the Evaluator Model (EM), to predict the stability score of a sequence of amino acids (Fig 2). This model was trained on a corpus of 107,948 proteins ("Corpus A") designed by experts using Rosetta [4] or dTER-Men [31] software (S5 and S6 Figs), and achieved high performance on held-out test data. Corpus A comprised the designs reported in [26], in [32], as well as previously unpublished designs.

The EM demonstrated predictive performance near the limit of the experimental data using a random testing subset. We reserved a set of 5000 randomly selected designs for testing model performance, with the remaining designs used for training. We calculated both the squared Pearson correlation coefficient (i.e. $r^2$) and the $R^2$ goodness-of-fit scores for the test set, for five versions of the EM (built by training from scratch five times). $R^2$ is more stringent than $r^2$ because it does not assume a linear regression of model predictions onto data, and can be negative when the model performs worse than simply predicting the sample mean. The mean $R^2$ score for the EM was 0.48 ($r^2 = 0.49$), slightly better than the relationships between trypsin and chymotrypsin stability scores (trypsin onto chymotrypsin $R^2 = 0.34$, chymotrypsin onto trypsin $R^2 = 0.38$; $r^2 = 0.47$). Ideally, trypsin and chymotrypsin stability scores would be equal; that EM predictions are as similar to stability scores as the per-protease scores are to each other suggests that the EM's performance is near the limit imposed by the assay.

We also evaluated EM performance when controlling for the possibility that it was memorizing artifacts of the design process rather than modeling underlying aspects of stability. It is

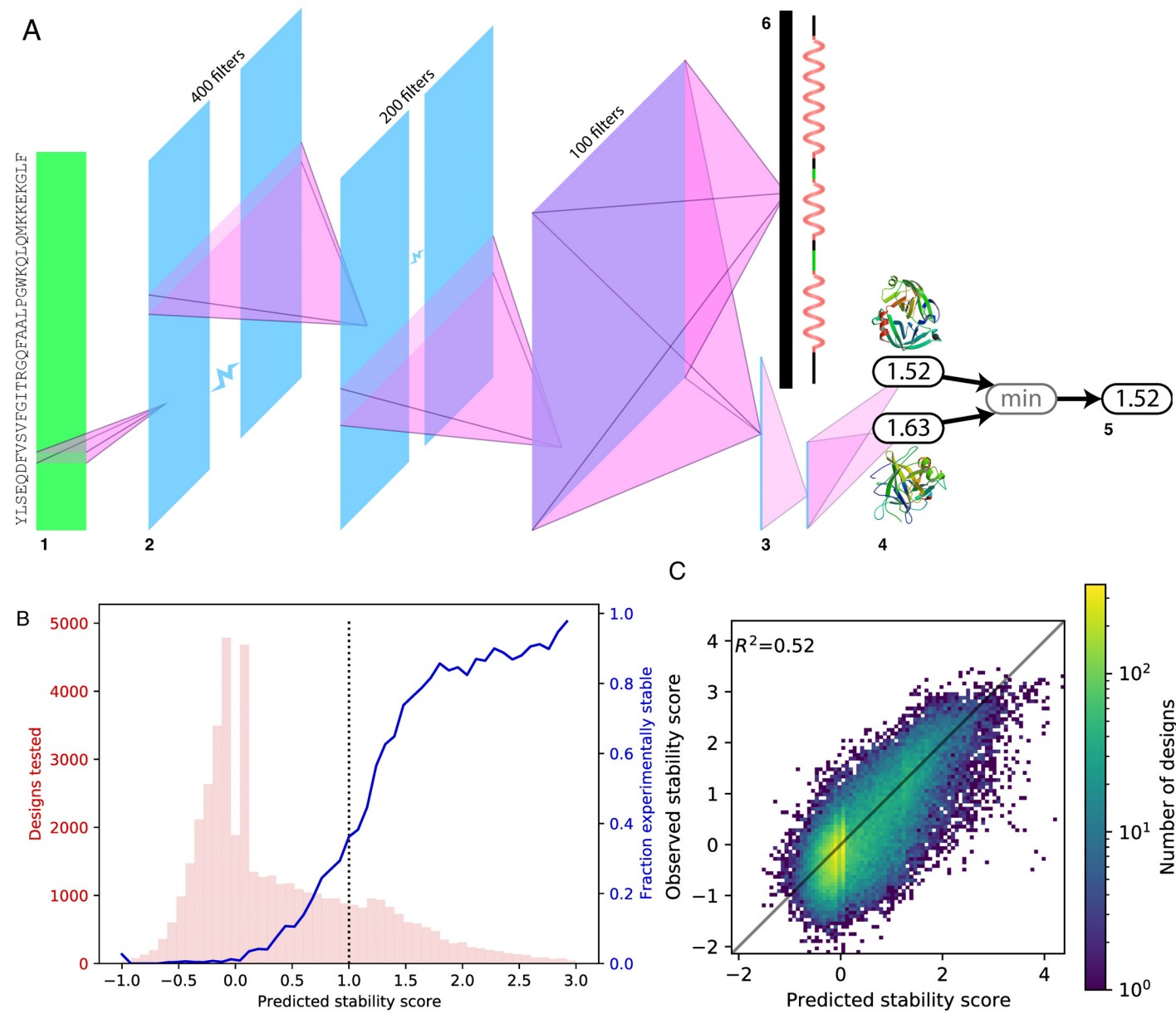

**Fig 2. Evaluator model and performance.** (**A**) Architecture of Evaluator Model. (**1**) Input: one-hot encoding of protein's primary sequence. (**2**) Three convolutional layers; the first flattens the one-hot encoding to a single dimension, successive filters span longer windows of sequence. Three dense layers (**3**) yield trypsin and chymotrypsin stability scores (**4**). The final stability score (**5**) is the minimum of the two. (**6**) A separate dense layer from the final convolution layer yields one-hot encoding of the protein's secondary structure. (**B**) Success of EM predictions on a library of new designs. We used the EM to predict the stability of 45,840 new protein sequences that the model had not seen before (described later as "Corpus B"); the distribution of predictions is shown in pink. The blue curve shows the fraction of these designs that were empirically stable (stability score >1.0) as a function of the model's a priori stability predictions (dotted black line: stability threshold for predicted stability). 281 outliers (predicted stability score <-1.0 or >3.0) excluded for clarity. (**C**) Predicted versus observed stability scores for the same data, with outliers included.

reasonable to think that a neural network might overfit the data by learning the patterns associated with a given topology or methodology. Topologies each have a loose but distinctive patterning of hydrophobic and hydrophilic residues [33]. Similarly, a particular design methodology may have a sequence-level "tell"—even one that is not easy for humans to discern [34]. In order to guard against overfitting to such features, we took advantage of the fact that

our training corpus comprised numerous topologies, design approaches, and sizes. We partitioned the corpus into 27 classes based on these factors. For each class, we tested performance on the designs within that class after training each of the three models on data from the other 26 classes. We again used $R^2$ goodness-of-fit scores, taking the mean of three retrained EMs for each held-out class. The EM achieved a mean $R^2$ of 0.17 over all classes, indicating that the previous results reflect some overfitting to topology or design methodology, but also demonstrating a substantial level of generalization. Performance on each held-out class is shown in S7 Fig. To verify that differences in topology and design yielded distinct proteins, we used BLAST with default parameters using BLOSUM62 [35] to identify each protein's nearest homolog from among the other 26 classes. Maximum bit scores and maximum percent identities were both low (mean ± SD of 34.5±5.1 and 47.0±8.3 respectively), indicating substantial differences between the 27 classes. Moreover, EM performance was not generally better for proteins that were more similar to those in other classes; we found negligible correlations between model error and bit score (Spearman $\rho = -0.0011$) and percent identity (Spearman $\rho = 0.014$).

We next asked whether the EM can predict thermal stability for short natural proteins. We procured from the literature the melting temperatures of single-site mutations of several proteins: Cold shock protein CspB [36], Regulatory protein rop [37], Pancreatic trypsin inhibitor [38], Transcriptional repressor arc [39], Toxin CcdB [40], $\alpha$-lactalbumin [41], and Subtilisin-chymotrypsin inhibitor-2A [42]. We ran these sequences through the EM to predict stability scores, and calculate the Pearson correlation coefficient between experimentally measured melting temperatures and antilogs of stability scores (Pearson $r$ because the units are not directly comparable, and antilogs because stability score is logarithmic) for each data set. Table 1 presents information about the data sets, the Pearson $r$, and associated p-values. Even without any training on natural protein thermal stability data, there are weak but consistently positive correlations between EM stability predictions and thermal stability.

We next sought to evaluate the EM's ability to predict the effects of sequence perturbations and to refine the stability of synthetic protein designs. To support this investigation, we constructed a new dataset ("Corpus B") of 96,799 designs, which fell into three main categories: expert designs from the held-out test subset of Corpus A subjected to a variety of perturbations, expert designs from the held-out test subset of Corpus A subjected to refinement, and novel neural network designs subjected to refinement (all described below). We used the EM to predict the stability of these new designs, and then tested them empirically in the stability assay. We analyzed the 45,840 of these designs for which the credible intervals for both measured protease $EC_{50}$ values were no greater than 2. Across this set of designs, the EM's predictions were highly consistent with the observed stability scores ($R^2 = 0.52$, $r^2 = 0.59$, Fig 2B and

**Table 1. Evaluator model performance on natural proteins.**

| Protein | N | Length | Pearson r | p-value |
|---|---|---|---|---|
| Cold shock protein CspB | 27 | 67 | 0.363 | 0.063 |
| Regulatory protein rop | 20 | 63 | 0.136 | 0.568 |
| Pancreatic trypsin inhibitor | 41 | 100 | 0.292 | 0.064 |
| Transcriptional repressor arc | 47 | 53 | 0.215 | 0.146 |
| Toxin CcdB | 78 | 101 | 0.186 | 0.102 |
| $\alpha$-lactalbumin | 24 | 142 | 0.380 | 0.067 |
| Subtilisin-chymotrypsin inhibitor-2A | 20 | 84 | 0.019 | 0.935 |

Seven data sets each contain experimentally measured thermal stability for a natural protein and $N - 1$ single site-mutations of that protein. We report the Pearson correlations between the antilogs of the predictions of the EM and the experimentally observed melting temperatures for each data set.

2C). We note the marked prevalence of predicted stability scores in two peaks slightly above and below 0; we attribute no meaning or significance to these peaks, however—a drawback of using neural networks, where interpretability is often elusive.

Many designs in Corpus B were expert designs mutated to test whether the EM could be misled by local or global changes to a sequence. Such mistakes could reveal weaknesses caused by the model's training data or approach. The EM was trained only on stability scores of expert designs created with stability as a goal (though it was exposed to natural protein sequences during training; see Methods), which made overfitting a particular concern. The EM might, for instance, learn to make predictions based mostly on a design's overall amino acid composition, taking advantage of statistical differences between the training set and the larger set of possible sequences. In that case, the EM may be relatively insensitive to single-site mutations or changes that alter the ordering of amino acids but maintain their overall frequencies. Conversely, if the EM has learned to make predictions based on specific memorized motifs present in the training data, then single-site mutations away from these motifs may cause the model to react unpredictably. To test these possibilities, we evaluated the effects of fourteen different types of perturbations to each of the 5000 designs in the original test set. These fell into three classes: single-site mutations (substitution, insertion, deletion), global rearrangements that did not change amino-acid frequency (sequence reversal, cyclic shifts), and global compositional changes (deleting half the amino acids).

The model was able to predict the stability of perturbed proteins with reasonable fidelity (S8 Fig). As expected, all fourteen classes of perturbations tended to decrease a protein's stability. Also as expected, the two classes of global disruptions tended to decrease stability more than the local mutations. To evaluate the EM's ability to predict decreases in stability, we limited analysis to cases where the base protein had, or was predicted to have, a stability score of at least 1. As seen in S9 Fig, the model systematically underestimated the impacts of these sequence perturbations. However, it ordered the fourteen perturbations reasonably well by mean destabilization (Spearman $\rho = 0.908$, $p = 7.30 \times 10^{-6}$) and showed some success at ranking the impacts of disruptions to individual proteins (Spearman $\rho = 0.432$, $p = 1.68 \times 10^{-202}$).

## Refining stability with a sequence-only model

Successfully predicting that mutations reduce stability could, in principle, be achieved by overfitting to signatures of expert design—mutated versions of such designs, lacking those signatures, could be judged as inferior. Therefore, we asked whether the EM can also predict stabilizing mutations, which presumably would not be seen by the model as improving upon an adventitious signature.

The EM was able to find multi-site mutations that increased stability score from among astronomically large sets of possible mutations to a collection of proteins. This demonstrates extrapolation of the EM's predictions to those regions of design space where a design's stability increases. More importantly, it raises the possibility of rapid and automatic improvement to existing protein design pipelines. To evaluate this possibility, we randomly selected a subset of 1215 of the test set designs, and subjected these sequences to incremental stabilization guided by the predictions of the EM. We performed a five-round beam search (Fig 3A) to generate successive single-site substitutions with higher predicted stability. Although this approach required the prediction of stability for hundreds of millions of primary sequences, the computationally efficient EM yielded these predictions in only a few hours using a single GPU. On average, this iterative refinement increased assayed stability more than ten-fold (i.e., one stability score unit) after five single-site substitutions (Fig 3B).

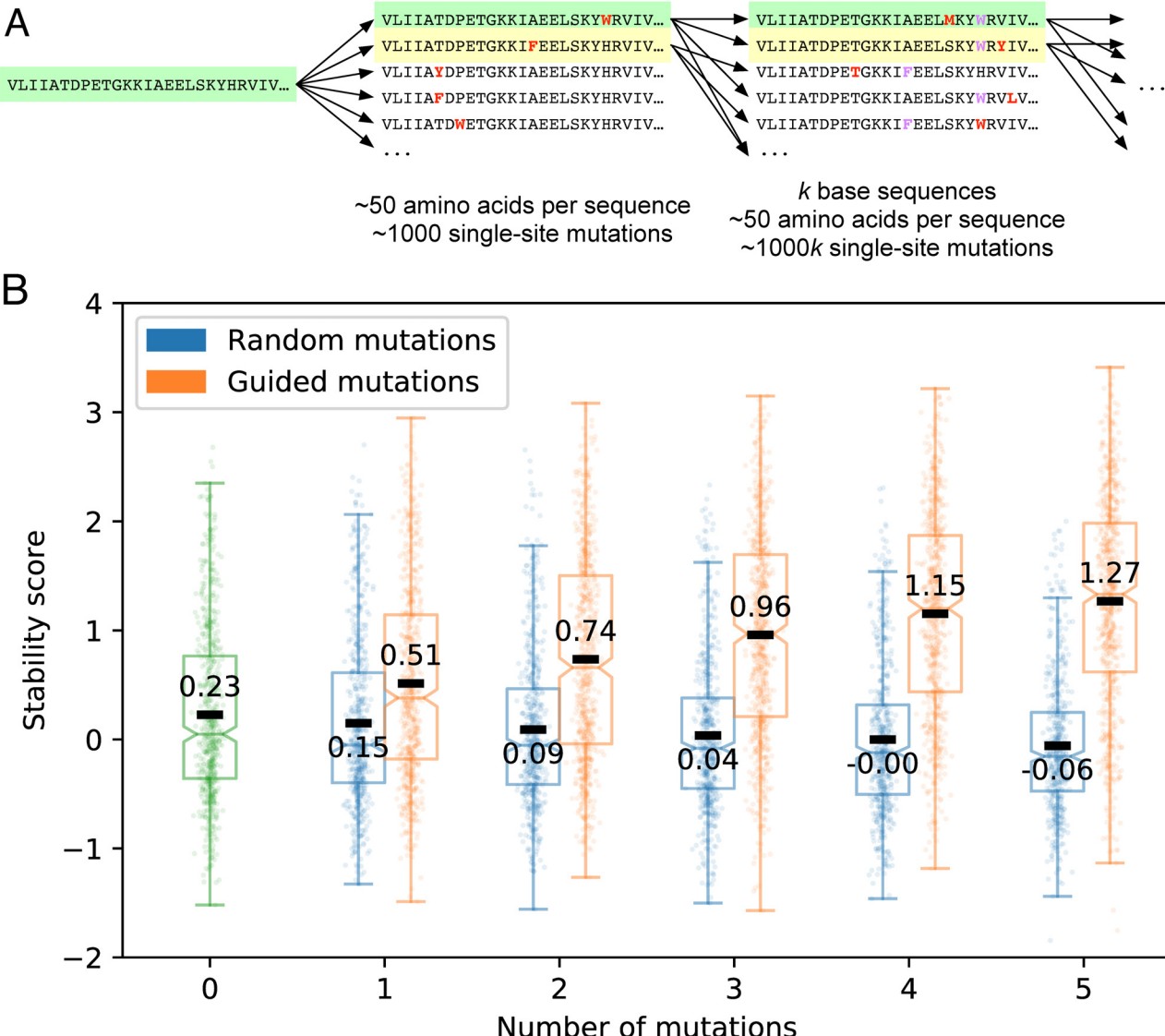

**Fig 3. Refinement and its effects.** (**A**) Beam search refinement. Refinement begins with a protein's amino acid sequence (left, green). All possible single-site substitutions are generated (bold red characters in middle sequences), and they are sorted according to the EM's prediction of their stability (middle). The design with the highest predicted stability (middle, green) is reserved as the product of refinement at this stage. The $k$ single-site substitutions with the highest predicted stability (middle, green and yellow; $k = 2$ in this illustration, though we used $k = 50$ to stabilize proteins) are then used as new bases. For each of the $k$ new bases, the process was repeated, combining all single-site substitutions of all $k$ new bases in the new sorted list (right). In this fashion, we predicted the best mutations of 1–5 amino acid substitutions for each of the base designs. (**B**) Effect of guided and random substitutions on expert-designed proteins. Guided substitutions (orange) raised the mean stability score from 0.23 in the base population (green) to 1.27 after five amino acid changes, as compared to random substitutions (blue) which dropped it to -0.06. Because stability score is logarithmic, the increase in stability is more than ten-fold after five guided substitutions. Annotated black bars indicate means, notches indicate bootstrapped 95% confidence intervals around the medians, boxes indicate upper and lower quartiles, and whiskers indicate 1.5 times the inter-quartile range.

Notably, this refinement is successful even though the EM's ability to accurately predict the effects of single-site mutations is limited. One of the proteins we subjected to guided refinement, "HEEH_rd3_0726", had single-site mutations exhaustively characterized previously [26]. The Pearson correlation coefficient between antilogs of predicted and experimental stability scores for this protein was only 0.419, not much higher than some of the correlations

seen with single-site mutations of natural proteins (Table 1). Yet, for this protein, three and four guided mutations increased stability 2.25-fold and 1.89-fold, respectively.

Substantial improvements to stability are seen even when we restrict analysis only to proteins that are already at least marginally stable and have $EC_{50}$ values well below the assay's ceiling (S10 Fig). This demonstrates that the EM can stabilize designs that are already somewhat stable, rather than only transforming entirely unstable sequences into partially folded structures. By limiting $EC_{50}$ values we ensure that the EM is not being misled by, or taking advantage of, unfolded-state predictions outside the assay's range.

Iterative stabilization of proteins also succeeds with respect to the individual components of the final stability score. In S11 and S12 Figs, we show stability scores for each protease separately. We see that the effect of guided substitutions is smaller, and the effect of random substitutions is larger, for trypsin than for chymotrypsin. As also seen in S4 Fig, trypsin stability scores tend to be lower than chymotrypsin stability scores, which means that they more often determine the final stability score. In every case, however, we see increasing divergence between guided and random substitutions as the number of substitutions increases.

To separate how much of the change in stability score was due to changes in protease resistance and how much was due to changes in predicted unfolded-state resistance, we examined the experimentally observed $EC_{50}$ values for each protease. S13 and S14 Figs show the results, broken down by protease. In both cases, there is increasing divergence between $EC_{50}$ values for guided versus random substitutions as the number of substitutions increases. Trypsin $EC_{50}$ values increase modestly as a result of EM guidance, while chymotrypsin $EC_{50}$ values hold steady. For both proteases, much of the increase in stability score is associated with predicted decreases in unfolded-state resistance, in the absence of a corresponding decrease in observed protease resistance. As with stability scores, we see the same pattern of results when we restrict analysis only to proteins that are already at least marginally stable and which demonstrate $EC_{50}$ values well below the assay's ceiling (S15 and S16 Figs).

## Generating and refining novel proteins with sequence-only models

Given that a data-driven neural network model can rapidly and successfully predict stability from sequence, we next asked if a similar model can also rapidly generate new stable designs. Specifically, we approached de novo protein design as a language generation task where the amino acids are the words of our language. We adapt deep learning sequence models from neural machine translation [43, 44] for our task. This architecture translates secondary structure sequences to primary sequences of the same length using an attention-based encoder-decoder recurrent neural network [45]. This yields primary sequences that are likely to have close to the requested secondary sequence and likely to be similar to sequences in the training set, with variation that likely reflects distributions in the training set. We refer to this model as the Generator Model (GM, Fig 4A). Beyond demonstrating proof of concept for this translation approach, we use the GM to create new proteins with which to evaluate EM-guided stabilization.

As a preliminary diagnostic to confirm that this class of model is fundamentally able to map between the domains of primary and secondary sequence, we consider the reverse task: predicting secondary sequence from primary sequence. We trained a model with architecture equivalent to the GM's to predict ground-truth secondary structure sequences derived from applying the DSSP algorithm [46] to Rosetta tertiary structure models for a set of primary sequences. This "Reverse GM", which recapitulates the functionality of tools like PSIPRED [47], achieves a secondary-sequence character error rate of 0.67% (CER: the average fraction of characters that are wrong; more precisely, the Levenshtein edit distance between the predicted and reference secondary sequences divided by sequence length). Even when evaluated

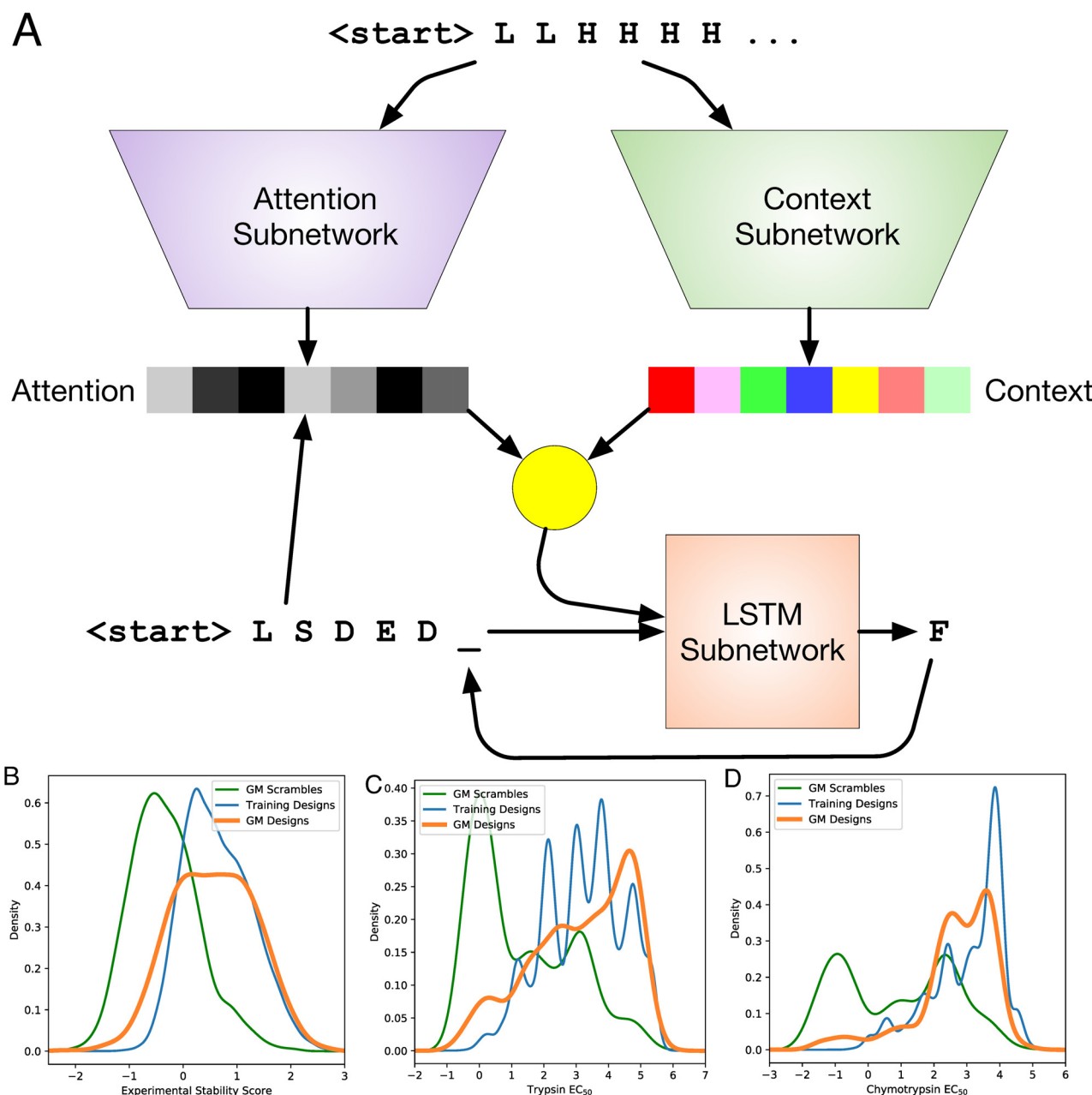

**Fig 4. Generator model and its performance.** (**A**) Architecture of the GM. Adapted for use with protein secondary and primary sequences from [45]. (**B**) Density plot of experimental stability scores for training designs, designs from the GM, and scrambles of the GM designs. (**C**) Density plot of trypsin $EC_{50}$ values. (**D**) Density plot of chymotrypsin $EC_{50}$ values.

on topologies that the model was not trained on, it achieves a CER of 9% for secondary sequence. We also used the Reverse GM to evaluate primary sequences produced by the GM, as described below.

We used the GM to create a set of primary sequences based on secondary sequences of expert-designed proteins that had been held out from the training set. To evaluate how well the model was able to learn principles of expert-designed proteins, we measured the perplexity [48, 49] of the predicted primary sequences relative to the actual primary sequences of the test

proteins. Perplexity, the standard metric in language modeling [50], is the antilog of the average of the per-residue cross entropy of the model's predictions. It can be intuitively understood as the average number of reasonable choices available to the model for each predicted amino acid, and can be as low as 1 when predictions are perfect. A perplexity of 19 would indicate chance (we exclude cysteine from the GM's vocabulary because disulfide bonds can invalidate the stability assay by preventing the fluorescent end of the expression vector from being washed away even when the protein has been cleaved [26]). As a control, we trained a basic recurrent neural network on the same data set. When tasked with predicting an amino acid in a protein's sequence based on the amino acids preceding it, this control model achieved a perplexity of 1.8. The GM expands upon such a model by considering, for each amino acid, not only the primary sequence generated so far but also the desired secondary sequence. In doing so, it achieves a perplexity of 1.1—by considering secondary structure information, it is able to produce primary sequences that are more reliably similar to those in the test set.

We selected 1000 GM designs with the highest agreement between the input secondary structure and the Reverse GM's predicted secondary structure, speculating that these designs might have underlying features best encoded by the GM. Note that this selection process further encourages sequences to be similar to those in the training set. These 1000 designs were assayed for stability as part of Corpus B. Their measured stability scores and $EC_{50}$ values were similar to those of the expert-designed proteins on which they were trained, and much higher than random scrambles of the same designs (Fig 4B–4D). We used BLAST [51] to compute the maximum identity of the designs with respect to designs in the training data set. Of the 993 designs that expressed at a detectable level, 275 had a maximum identity less than 95% and 47 of these 275 (17%) had an empirical stability score greater than 1.0 (classified as "stable"). Of the 120 designs with a maximum identity below 75%, 17 (14%) were stable. The GM produced stable sequences for six of the nine topologies that were represented by at least 500 designs in the training set. Note that the GM is simply learning to translate from secondary sequences to primary sequences; it knows nothing explicit about tertiary structure or stability. Because we wanted to generate stable sequences, the training set for the GM was only a subset of Corpus A, with many unstable designs excluded and greater weight given during training to stable designs (see Materials and methods). The GM was able to nearly match the stability distribution of this stability-enriched training set (Fig 4B).

We questioned whether EM-based refinement would improve these 1000 GM designs as it did expert-designed proteins, or, alternatively, whether the EM would see designs by another neural network model trained on the same data as unimprovable. Guided substitutions overall yielded ten-fold increases in the stability of GM designs (Fig 5A, stability score increase of 1.0). By contrast, random substitutions overall yield proteins that are less stable than those created by the GM. Importantly, the increase in stability with EM refinement is apparent even for proteins that are the most different from those in the training corpus (Fig 5B, S17 Fig). This demonstrates that we are able to design and iteratively improve highly stable proteins de novo, using an automated, rapid pipeline.

S18 and S19 Figs show increasing divergence between guided and random substitutions for each protease's individual stability score as the number of substitutions increases. S20 and S21 Figs show increased divergence between guided and random substitutions for each protease's $EC_{50}$ as the number of substitutions increases.

## Guided refinement demonstrates biases but yields stable monomers

We observed that individual single-site substitutions were much more successful at improving assayed stability when guided by the EM than when carried out at random. Considering

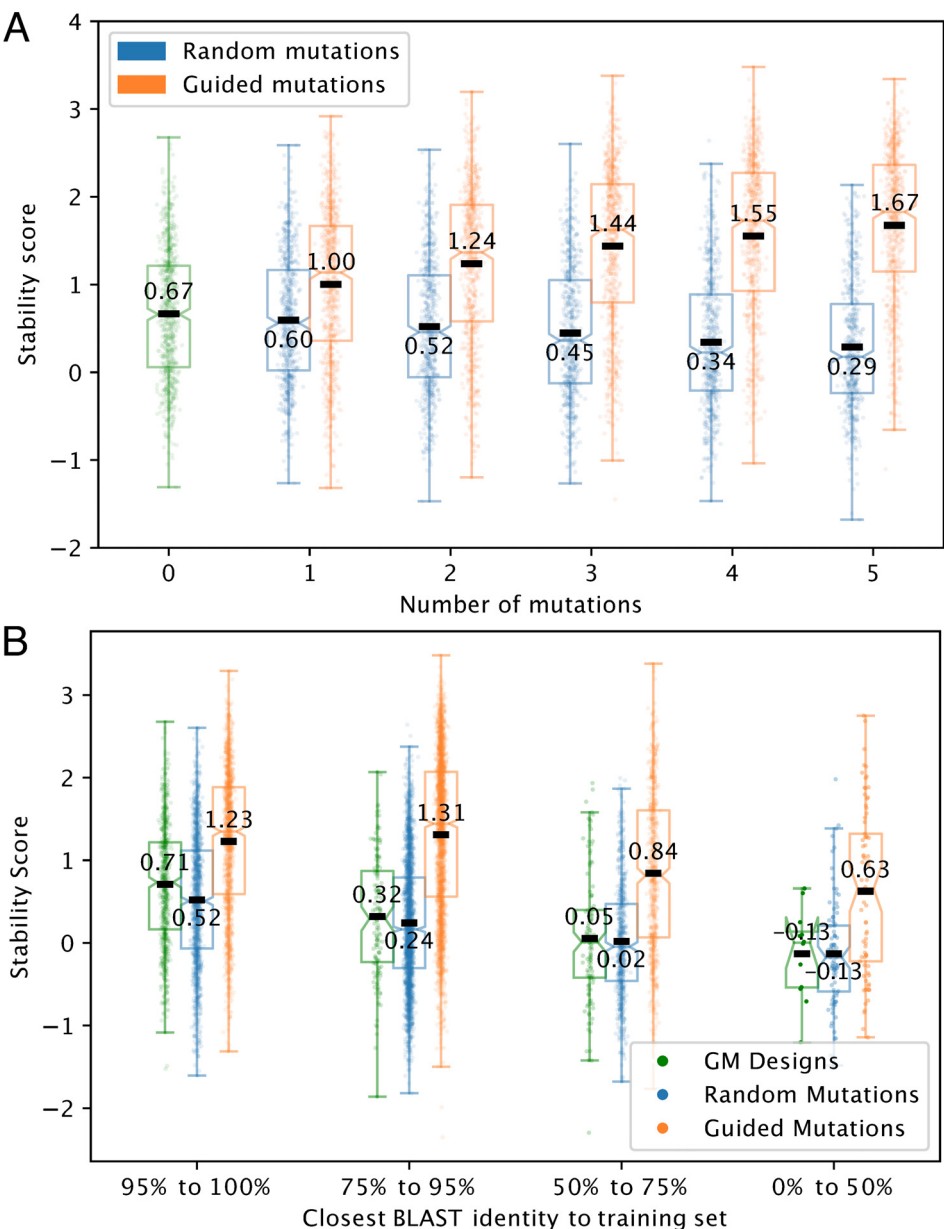

**Fig 5. Refinement of GM designs, overall and as a function of novelty.** (**A**) Effect of guided and random substitutions on designs created by the GM. The base stability score was much higher for this population of designs than for the expert-designed proteins tested, with a mean of 0.67; EM-guided refinement further increased it to 1.67. As with the expert-designed proteins, this demonstrates a ten-fold increase in stability. Random substitutions again had a deleterious effect, dropping mean stability to 0.29. (**B**) Stability of GM designs, and guided and random substitutions within those designs, as novelty increases. We consider designs to be more novel when BLAST percent identity with the most-similar design in the training corpus is lower.

mutations to both expert-designed and GM sequences, there were 5850 guided single-site substitutions evaluated in the Corpus B, and 4798 random substitutions. For each type of substitution (i.e. for each change from amino acid *x* to amino acid *y*) we calculated the mean stability change when that substitution was applied due to the EM's guidance and when it was applied at random. There were 191 of these types of substitution for which we have data in both

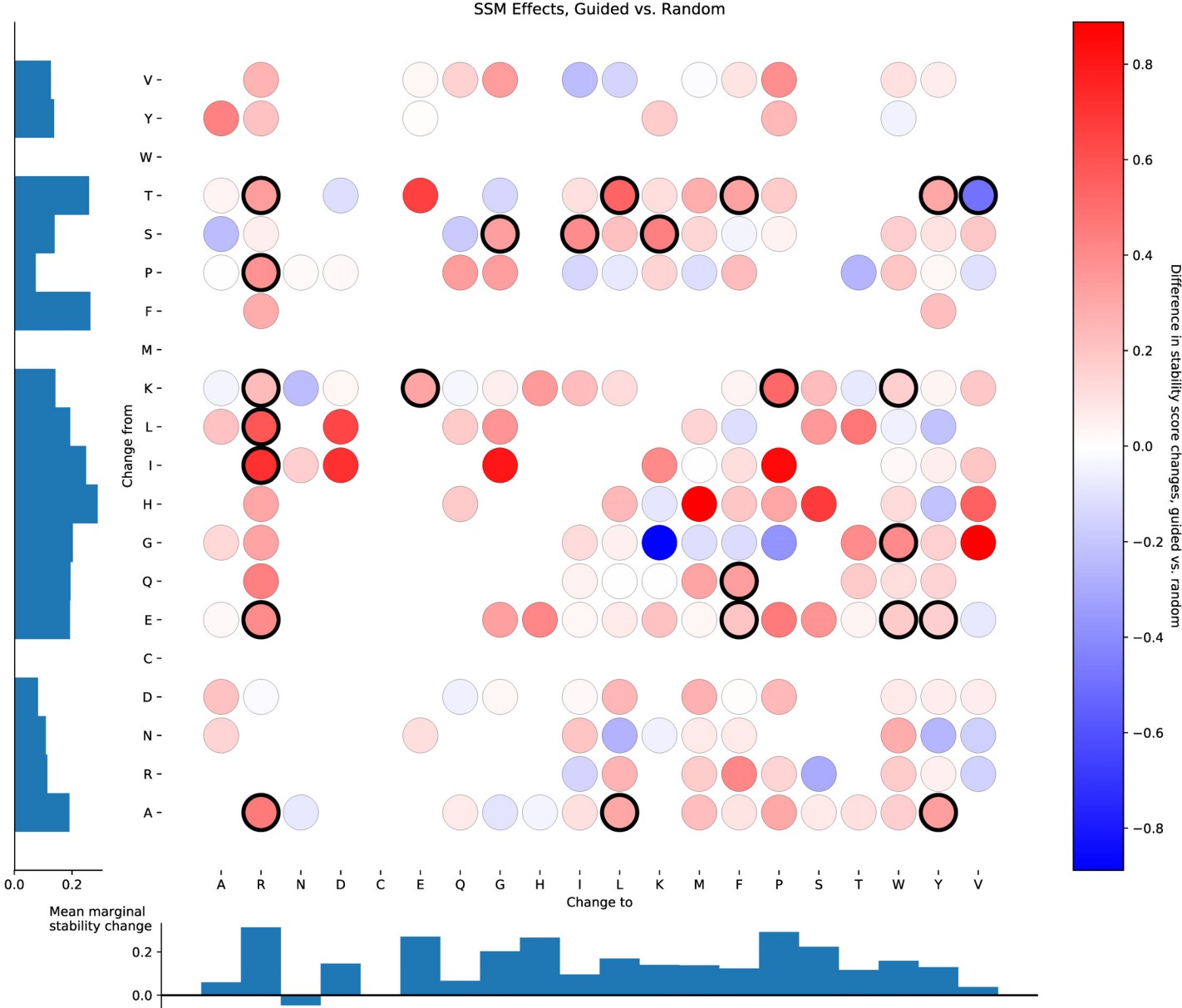

**Fig 6. Differential effects on stability between guided and random single-site substitutions.** For each original amino acid (indexed on the y-axis) and each replacement amino acid (indexed on the x-axis), the mean effect on stability when that substitution was guided by the EM is computed, as is the mean effect on stability when that substitution was applied randomly. The difference between these two effects is plotted for each from-to pair that was represented in the data; redder circles indicate that guided substitutions were more beneficial for stability, bluer circles indicate that random substitutions were more beneficial. Circles with heavy black outlines showed a significant difference (two-sample unpaired two-tailed t-test, $p < 0.05$ uncorrected) between guided and random effects. Bar graphs indicate mean differences in stability score (guided substitutions minus random substitutions) averaged across all replacement amino acids for each original amino acid (left) and and averaged across all original amino acids for each replacement amino acid (bottom).

conditions. Of these, guided substitutions were more successful than random for 148 of them, with an average stability increase over all types of 0.155 (Fig 6). As a somewhat arbitrary threshold for considering differences between guided and random substitutions to be reliable, we ran t-tests between the two approaches for each type of substitution (two-sample unpaired two-tailed t-test, $p < 0.05$ uncorrected for multiple comparisons). 24 of the substitution types

showed significant differences between guided and random substitutions, indicating sufficient examples of both treatments as well as sufficient differences between them; in 23 of those, guided substitutions were more successful than random.

Further breaking down these analyses and looking at guided and random substitutions separately, we see that guided substitutions increased stability score 4138 out of 5850 times (70.7%); the mean stability change over all guided substitutions was 0.208 (S22 Fig). Out of 196 types of guided substitution, 43 showed a significant positive effect on stability (two-tailed t-test, $p < 0.05$ uncorrected for multiple comparisons), and 2 showed a significant negative effect. If guided substitutions had no real effect on stability, we would expect to see approximately 10 types crossing this significance threshold in each direction. Random substitutions, meanwhile, decreased stability score 2731 out of 4798 times (56.9%); the mean stability change over all random substitutions was -0.074 (S23 Fig). Out of 318 types of random substitution, 6 were significantly helpful (two-tailed t-test, $p < 0.05$ uncorrected), and 55 were significantly deleterious; we would expect approximately 16 in each direction, if random substitutions had no real effect on stability. The general lack of improvement with random substitutions conforms with the intuition that stabilizing proteins is difficult. That EM-guided substitutions perform so well is encouraging but invites further scrutiny.

The EM's refinement of proteins frequently substituted amino acids with tryptophan (S22 Fig)—it was the amino acid of choice in 3276 of the 5850 guided single-site substitutions. This raises the possibility that the EM has learned how to exploit gaps between what the stability assay is actually computing (a difference between observed protease resistance and predicted unfolded-state resistance) and the assay's objective (protein stability). For example, indiscriminately adding tryptophan to a sequence might cause proteins to aggregate or undergo nonspecific hydrophobic collapse [52], or even to form regions of stable structure [53]. Either of these could increase protease resistance without necessarily increasing the stability of the intended tertiary structure. However, random mutations to tryptophan show virtually no change in assayed stability score (mean increase of 0.016), compared to a substantial increase in stability score when those substitutions are guided by the EM (mean increase of 0.215). Lysine and glutamic acid are the two most common residues to be mutated to tryptophan, and these residues usually occur on the surface of the protein. By favoring these mutations, the EM could be implicitly learning to bias tryptophan substitutions at surface sites, versus random mutations which would be more uniformly distributed and thus less likely to lead to aggregation. However, if those biases were leading to aggregation and thus artificially inflated stability score, random mutations from lysine or glutamic acid to tryptophan should show a similar increase in stability score as guided mutations. Guided mutations, however, increase stability scores substantially more. Taken together, these observations show that increasing the tryptophan count of a protein in and of itself does not increase stability—the changes to tryptophan must be at the right location for the protein in question.

To directly test the quality of the proteins generated, we selected by hand—on the basis of low sequence identity to the training set, diversity of primary and secondary sequences, and the progenitor sequences for four apparently stable refinements—twelve designs for further analysis. We expressed these designs in *Escherichia coli* with N-terminal His6 affinity purification tags and purified them using immobilized metal affinity chromatography (IMAC, Fig 7A). Eight of the designs were expressed as reasonably soluble proteins and seven of those were monomeric as judged by size exclusion chromatography (SEC). Mass-spectroscopy confirmed that the purified fractions were predominantly molecules of the correct molecular weight. Circular dichroism (CD) spectroscopy with wavelength scans and thermal melts were used to assess the secondary structure and overall stability, respectively, of each of the soluble proteins. The nmt_0457_guided_03 and nmt_0994_guided_02 proteins, expected to be all

| A Name | Max % Ident. | Topology | Stab. Score | Solu. | SEC | CD Scan | CD $T_m$ |
|---|---|---|---|---|---|---|---|
| nmt_0282 | 43 | HHEE/HHEEE | 0.60 | – | agg | n/a | n/a |
| nmt_0282_guided_05 | 48 | HEE/HHEEE | **2.69** | – | agg | n/a | n/a |
| nmt_0303 | 48 | EHEEH/EHEEHE | 0.01 | +++ | mon+agg | folded | >80°C |
| nmt_0303_guided_04 | 44 | EHEEH/EHEEHE | **1.29** | +++ | mon | folded | >99°C |
| nmt_0861 | 56 | H/EEH | -0.51 | + | mon | unfolded | n/a |
| nmt_0861_guided_04 | 49 | EHH/EEEH | -0.12 | + | mon | borderline | n/a |
| nmt_0990 | 40 | HHEH/HHH | -1.20 | ++ | mon | unfolded | n/a |
| nmt_0990_guided_05 | 38 | EHHEH | **1.25** | +++ | oligomer | folded | >99°C |
| nmt_0315_guided_04 | 45 | EHEEHE | **1.88** | – | agg | n/a | n/a |
| nmt_0457_guided_03 | 48 | HHHH | **2.00** | +++ | mon | folded | >99°C |
| nmt_0752_guided_05 | 37 | EEH | -0.04 | – | agg | n/a | n/a |
| nmt_0994_guided_02 | 46 | HHH/HHHH | **1.88** | ++ | mon | folded | >99°C |

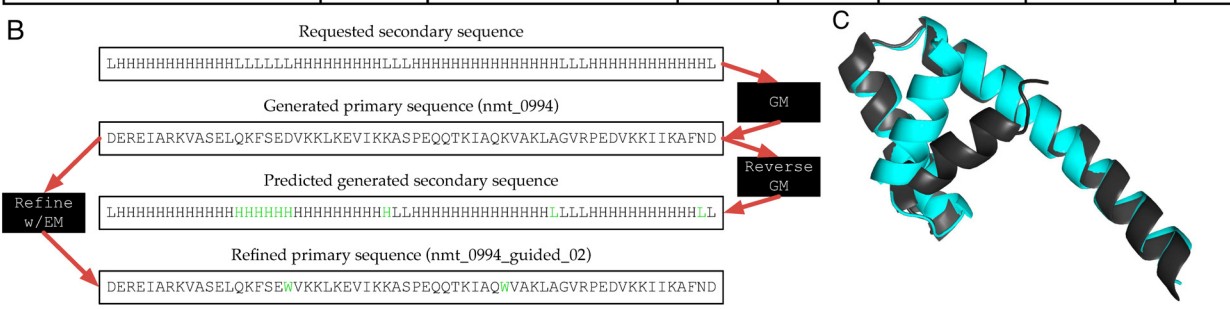

**Fig 7. Laboratory analyses of GM proteins.** (**A**) Results of targeted analyses of twelve GM proteins. All twelve proteins had less than 60% identity with respect to the entire set of training proteins, as calculated by BLAST. Reported topology was predicted by PSIPRED [47] and Rosetta (in that order, when predictions differ). (**B**) "Life cycle" of one refined protein, nmt_0994_guided_02. The design began with a requested secondary structure fed into the GM. The GM produced a primary sequence (nmt_0994) stochastically translated from that secondary structure; however, the Reverse GM correctly predicted that two of the requested helices were actually merged into one in the generated protein's structure. EM-guided refinement then changed two residues to tryptophan, which raised the empirical stability score from -0.18 to 1.88. Green characters highlight differences from original sequences. (**C**) Crystal structure for nmt_0994_guided_02 (dark grey), showing that it also has the three helices predicted by the Reverse GM for its pre-refinement progenitor. It is shown aligned to the structure predicted by AlphaFold2 (cyan). The prediction and the crystal structure have a $C_\alpha$RMSD of 3.4 Å.

alpha helical, show distinct minima at 222nm and 208nm, consistent with the intended helical structure. For all the other designs, the spectra are consistent with a mixed alpha-beta secondary structure (S24 Fig). Thermal stability of the designed proteins was determined by monitoring the CD signal at 222nm as a function of temperature. Five of the eight monomeric/oligomeric designs are highly thermostable (S24 Fig); four do not unfold even at a temperature of 99°C. These four are precisely the non-aggregating proteins with stability score > 1, providing additional experimental validation to the previously published finding [26] that stability scores are reasonably correlated with thermal stability. For one of these highly stable designs (Fig 7B) we obtained a crystal structure (Fig 7C); this showed an all-helical structure,

consistent with predictions of the Reverse GM, CD, and AlphaFold2 [54]. Two of the three helical segments observed in the crystal structure match AlphaFold2's prediction, with the third helix taking a slightly different trajectory from the prediction. We also predicted the tertiary structure of all twelve designs using Rosetta (S25 Fig). These results confirm that our data-driven pipeline can design stable monomeric proteins, amenable to analysis with current tertiary structure prediction models, without itself making any explicit calculation of tertiary structure.

## Refinement under constraint

To further examine how EM-guided refinement increases stability score, as well as to demonstrate the feasibility of performing this refinement under constraints of varying strictness, we generated and assayed a third dataset ("Corpus C"). This consisted of 97,702 proteins, including three sets representing refinements governed by increasingly stringent constraints. All GM designs on this chip were required to have less than 50% sequence identity with the most similar expert-designed protein, and all refinements used a new version of the EM trained on Corpora A and B.

The first set of designs demonstrated that refinement could be successful without decreasing USM predicted $EC_{50}$ values for either protease. This greatly reduces the potential for the EM to exploit weaknesses in the USM. Refinement of expert-designed sequences under this constraint showed a slight improvement in stability, compared with a substantial decrease to stability for random substitutions (which were permitted to decrease USM predictions). After four EM-guided substitutions, the mean stability score for this set of refined proteins was 0.6 stability score units higher than for random substitutions (S26 Fig). In comparison, initial GM designs were on average much less stable than expert designs, leaving random substitutions with very little room to decrease stability. However, in contrast to expert designs under these constraints, these GM designs were substantially stabilized. After four guided mutations, they were 0.51 stability score units above random substitutions (S27 Fig).

The second set of designs showed that refinement could be successful when a no-tryptophan constraint was added to the previous USM constraint. In Corpus B, refinements were dominated by substitutions that replaced other amino acids with tryptophan. As described above, we were concerned that these substitutions may be artificially inflating stability scores. Notably, this predilection for tryptophan was less apparent even in the least-constrained set of refinements in Corpus C (S28 Fig), likely because mutations to tryptophan tended to decrease USM $EC_{50}$ predictions. Explicitly prohibiting tryptophan eliminated the 823 mutations to tryptophan seen in the least-constrained refinements, but otherwise altered the pattern of changes and their effects relatively little (S29 Fig). Due to space constraints on this library's chip, we did not evaluate random no-tryptophan mutations of expert-designed proteins. However, guided refinement of expert designs yielded maintenance or slight improvement in stability (S30 Fig), comparable to what was seen when mutations to tryptophan were permitted. GM designs refined without tryptophan also responded similarly to refinements where it was permitted, with four guided mutations ending up at 0.56 stability score units higher than four random mutations (S31 Fig).

The third set of designs in Corpus C demonstrated that stability score could be maintained or only slightly decreased when substitutions were made under constraints that expanded to exclude a set of nonpolar amino acids: alanine, phenylalanine, isoleucine, leucine, methionine, valine, tryptophan, and tyrosine. Even with this stringent constraint, GM designs did not decrease in stability score (S32 Fig), and expert designs decreased only 0.22 stability score units after four mutations (S33 Fig). Under such heavy constraints, EM-guided refinement

most often selected lysines to change into other amino acids, most frequently (and most reliably successfully) aspartic acid and glutamic acid (S34 Fig).

The benefits of EM-guided refinement are not universally effective across arbitrary amino acid sequences. We attempted to refine a set of randomly scrambled expert-designed proteins, with no success. This was true regardless of whether all mutations other than those which decreased USM $EC_{50}$ predictions were permitted (S35 Fig), tryptophans were prohibited (S36 Fig), or the larger set of nonpolar amino acids were prohibited (S37 Fig). This failure to refine scrambles provides evidence that refinement is not merely changing arbitrary sequences in such a way that they fool the assay into viewing the sequences as more stable. The failure to refine scrambles contrasts notably with the success at refining GM designs even when they are not initially very stable—there may be some degree of baseline plausibility to the unstable GM designs, which random scrambles do not possess, that permits successful refinement.

## Discussion

Here we demonstrate that a convolutional neural network model (the EM) can predict the stability—measured using a yeast display assay that permits evaluation of 100,000 proteins per experiment—of novel mini-proteins given only primary sequences as input. Specifically, when testing on held-out designs derived from the same datasets used in training, the EM predicts stability with high accuracy, close to the ceiling imposed by the noisiness of the assay. This shows that the EM can readily identify patterns that relate designed protein sequences to their stability. Its generalization to new classes of proteins (with a different size, topology, or design method) is not as good, which could be due to limited training data (the designs only explore a tiny sliver of protein space) or limitations of a sequence-based model that does not explicitly consider the protein's 3D structure. However, the EM's predictions are still generally better than predicting the mean stability score of the novel class of proteins, already a non-trivial achievement.

To explore additional regions of design space beyond the training data, we created a large set of proteins whose sequences were manipulated in various ways, ranging from single-site mutations to global disruptions. These manipulations were designed to probe anticipated weaknesses of sequence-based models, and many of the manipulations were not represented in the training set. The EM's predictions systematically underestimated the magnitude of mutational impacts, exposing limitations of the model. However, it successfully predicted that most mutations were deleterious and successfully ranked different kinds of manipulations by effect size, providing further evidence of the EM's ability to generalize to new proteins. In addition to quantifying the EM's ability to predict such disruptions, we hope that this set of data can serve as a benchmark for other sequence-based models of protein stability in the future.

The yeast display stability assay has drawbacks. There is relatively high experimental noise, the results depend on multiple assumptions and models, and stability values are limited by the protease resistance of the expression vector. The tradeoff, however, is incredible throughput at relatively low cost. Given that machine learning models tend to perform (and in particular to generalize) better with a high quantity of noisy data than with a low quantity of cleaner data, this assay is a good partner for machine learning methods. Although we are cognizant of the possibility that the EM is learning to exploit weaknesses, in the USM or elsewhere in the approach, we are reassured by numerous control analyses. Moreover, we have demonstrated through empirical laboratory experiments that the process can yield soluble, monomeric, and stable designs.

We show that over a few substitutions using a model-guided beam search, proteins are substantially stabilized relative to the starting point, and even more so relative to an equal number

of random substitutions. In addition to being accurate, the EM is orders of magnitude faster computationally than physics-based methods (such as [55]), with the ability to generate on the order of 100,000,000 predictions per GPU-hour. This speed means it can readily be incorporated into existing design approaches, as a complement to more computationally intensive physics-based tools (e.g. [56]). Historically, optimization of protein properties has often been achieved through directed evolution; recently, machine learning and other computational techniques have been integrated to accelerate convergence [57]. The EM could also prove useful in such a context.

Recent studies using language models to analyze or generate proteins have been promising, demonstrating an ability to generate realistic looking amino acid sequences, sometimes guided by desired structural or functional properties, after training on large sets of natural proteins. We demonstrate that the GM can translate an input secondary structure into a primary sequence that achieves a stable tertiary structure. This approach is able to generate hundreds of thousands of designs, of comparable stability to those generated by experts using Rosetta, in a day. It is also possible to sacrifice stability performance for novelty, generating marginally stable but highly distinct proteins (with < 50% amino acid identity to their nearest neighbors in the training set) that can then be stabilized through EM-guided refinement. This success provides evidence that a combination of imperfect models can still lead to remarkable generalization into new areas of design space. Encouragingly, upon experimentally determining the crystal structure of one of these designs, we found that the designed secondary structure (as predicted by the Reverse GM) closely matched the real structure.

High-throughput cycles of design-test-learn are becoming common in protein design [26, 58]. Here, we show that a neural network can be a powerful tool for learning patterns that relate a protein's sequence to a desired property (in this case stability). Further, we show that neural network models can inform subsequent rounds of design, either by refining sequences from previous rounds, or by designing completely new ones. This can succeed even when models are overfitted or otherwise generalize imperfectly, as long as they carry a reasonable amount of information about the domain. Overall, this study provides a framework and grounding for using sequence-based neural networks in other design-test-learn contexts. These could include optimization targets beyond stability, such as protein binding.

Although the EM and GM do not predict the 3D structure of the design, there have recently been large advances in the accuracy of machine learning-based computational models for structure prediction [8, 9, 54, 59]. We found that one such model (trRosetta) could predict the crystal structure that we obtained for one of our designs with reasonable accuracy ($C_\alpha$RMSD of 3.7 Å), including correctly predicting the secondary structure and the relative orientation of secondary-structure elements, despite differences in finer details. Machine learning-based models can predict tertiary structure fairly quickly. Going forward, linking these approaches together could be used to rapidly generate 3D structural models for entire libraries of the most promising designs generated by the GM and refined by the EM, which could in turn be used as scaffold libraries for various design applications. A separate line of inquiry is whether the most recent models for tertiary structure could be used to directly predict protein stability. Perhaps variability in the predictions made, or deviations between their predictions of secondary structure and those of more traditional secondary structure models, might correlate negatively with stability. The slower prediction time of current tertiary structure models (hours, versus sub-millisecond) would make this impractical for very large libraries, however.

While the current EM is limited by both architecture and training data to small proteins, it may be possible to extend its capabilities to successfully accommodate larger proteins. This would require one of several possible architectural changes, but more significantly would likely require stability data for a large number of larger proteins. The stability assay used here fails to

correlate with $\Delta G_{unfolding}$ for at least some larger proteins [60], so another approach to generating large-scale stability data might be needed. Such an extended EM might have useful applications in medicine. Recent mRNA vaccines against SARS-CoV-2 [61, 62] have used a stabilized protein to elicit an immune response to the virus's spike protein in its prefusion configuration. This construct [63] depended on techniques discovered and developed earlier [64] for a related coronavirus. Discovering how to stabilize viral proteins in this manner has historically required a great deal of laboratory experimentation; an extended EM might be able to help speed such discoveries, by ranking or even suggesting the most promising mutants to investigate. Even without extending the EM to accommodate larger proteins, there may be medical utility: increasing the stability of small biologics benefits their shelf life, resistance to degradation in the body, and production efficiency. EM-guided stabilization also might be able to provide more stable bases for the evolution of proteins with new functions [65].

By combining the EM and the GM, we achieve a data-driven pipeline that enables straightforward extension into new regions of protein design space, and can be used in optimization tasks in which constraints are placed on the changes made. We demonstrated success at refinement while preventing USM predictions from increasing, while excluding tryptophan, and while excluding a large set of nonpolar residues. Other potential constraints include optimizing stability without increasing overall hydrophobicity, without disrupting a motif responsible for binding, without changing predicted secondary structure, or while maintaining a complementary tool's prediction of tertiary structure or charge distribution within certain bounds. This validated pipeline opens the door to the engineering of new kinds of small proteins.

## Materials and methods

We initially trained the EM on three Libraries of mini-proteins comprising Corpus A, each reflecting different design methodologies protein sizes, and topologies, listed in S5 Fig. Libraries consist of primary sequences, secondary sequences predicted by the DSSP algorithm [46], and experimentally measured stability scores, as quantified using the high-throughput assay from [26], with modifications described below. As indicated by the table, some data are from other publications, while some are being published as part of this study. Experimental stability score data for the "Rocklin" data are aggregated from all four design rounds from [26]. For Corpora B and C, experimental data come from independent experiments conducted as part of this study.

In cross-validation of the EM, we further subdivided Libraries by protein topology. We grouped designs in this way to ensure that different cross-validated groups did not include designs that shared the same topology, sequence length, and design protocol. S5 Fig maps each dataset into one of these Libraries. S6 Fig shows experimental stability scores for designs and control sequences from each Library, or groups of designs within a single Library (described below), broken down by protein topology. This figure also shows the distribution of protein length and number of designs and controls in each category.

Library 1 consists of all designs from all four design rounds from [26]. We recomputed stability scores for both designs and controls using the new USM developed in this study (S6A Fig). The control sequences ("scrambles" in S6 Fig) include all three types of controls from [26]: fully scrambled sequences (made by completely randomizing the designed sequence), "patterned" scrambles (made by randomizing the designed sequence, while preserving the patterning of hydrophobic and hydrophilic amino acids, as well as the positioning of glycines and prolines), and designed sequences with an aspartic acid mutation introduced at a buried hydrophobic residue. Recomputed stability scores for designs and controls mirror the trends observed with the original stability scores in [26]: most controls have stability scores near zero,

designs tend to have higher stability scores than controls, and the fraction of stable proteins (stability score > 1) is higher for designs than it is for controls. These trends qualitatively hold across all four topologies, which are named according to the ordering of secondary-structural elements in the design (H indicates a helix and E indicates a strand).

Library 2 consists of designs from [32], which span several topologies (4h: four-helix bundles; HHH: three-helix bundles; beta_grasp: beta-grasp fold; coil: helical bundle; ferredoxin: ferredoxin fold; thio: thioredoxin fold; fold2 and fold4: new folds from [32]). S6B Fig shows stability scores for these designs, along with scrambled controls generated from a subset of designs by randomly shuffling the entire designed sequence. As expected, most scrambles have stability scores near zero. For some topologies, designs tend to have higher stability scores than scrambles, indicating that their stability arises from more than just the overall amino-acid composition of the sequence.

Library 3 consists of three groups of designs, each of which are being published as part of this study. The first group are "blueprint-based" designs made using Rosetta. We designed these proteins using the same protocol as [26], except that we designed the proteins to be longer: 64–65 amino acids in length instead of 40 amino acids. We did so by modifying the "blueprints" from [26] to encode for longer proteins, while preserving the overall topology. We did this for three of the four topologies from [26]: EHEE, EEHEE, and HEEH. S6C Fig shows stability scores for these designs and scrambled controls made from a subset of designs. All scrambled controls in Library 3 were "patterned" scrambles, as described above. Designs tend to have higher stability scores than scrambles, suggesting that their stability arises from more than just amino-acid composition or the patternings described above.

The second group of designs from Library 3 were designed using dTERMen [31]. The detailed design protocol is described in [66]. Briefly, the protocol involved identifying sets of tertiary structural motifs that could be assembled into compact mini-proteins, connecting these motifs with loops so that they formed single chains, and then using dTERMen to design amino-acid sequences onto those single-chain backbones. This strategy was used to generate proteins spanning several topologies, as described in S6D Fig. HHH refers to three-helix bundles. All other topologies are named using the pattern "$XaYb$", where $X$ indicates the number of helices and $Y$ indicates the number of strands in the design. S6D Fig shows experimental stability scores for these designs, along with "patterned" scrambled controls. As above, for many of the topologies, designs tend to be more stable than the scrambled controls.

The third group of designs from Library 3 are Rosetta designs that were designed in a similar manner to the dTERMen designs. Instead of using dTERMen to connect motifs with loops and design amino-acid sequences onto backbones, however, we used Rosetta to perform these tasks. As shown in S6E Fig, this protocol was less successful than the dTERMen protocol: both designs and the "patterned" scrambled controls from this group tended to have stability scores near zero, and less than 1% of designs achieved stability scores greater than 1.

## Stability score

We experimentally measured stability scores using the high-throughput assay described in [26]. We independently assayed each dataset of mini-proteins used to train the EM (see S5 Fig), with the exception of the "Rocklin" dataset, for which we used published data from [26]). For the "Longxing" dataset, we followed the same method as [26]. For the other datasets used to train the EM, we used a modified version of the method described in [67], and conducted these experiments at the University of Washington BIOFAB (http://www.uwbiofab.org/) using Aquarium software (https://www.aquarium.bio/) to automate the workflows (https://github.

com/aquariumbio/yeast-display). For all training datasets, we used the original yeast-display vector from [26].

We assayed Corpora B and C at the BIOFAB using the modified method from [67]. However, in these experiments, we also used the modified version of the yeast-display vector from [67], which was evolved to have increased resistance to trypsin and chymotrypsin. This evolved vector expands the dynamic range of the assay. In order to compare stability scores collected with the original and evolved vectors, the BIOFAB assayed the round 4 library from [26] using the modified method and evolved vector. In the section "Reconciling multiple protein libraries" we describe how we use the resulting data to transform stability scores obtained using the evolved vector into values comparable with those obtained using the original vector.

### Unfolded-state model

Stability scores, yielded by the protein stability assay [26], are defined as the difference of two quantities: (1) empirical resistance to proteolysis, expressed in $EC_{50}$, and (2) an estimate of the protein's resistance to proteolysis if it were always in an unfolded state, as a function of its particular sequence of amino acids. The goal is to factor out baseline proteolysis resistance, leaving only resistance due to being in a folded configuration. The estimate of a sequence's resistance when unfolded comes from the USM, which is trained on $EC_{50}$ data generated for scrambled sequences that are highly unlikely to be stable. The USM reported in [26] appeared to be successful at generating predictions that were more useful than raw $EC_{50}$ values. However, during refinement, the EM might theoretically exploit imperfections in the unfolded-state model to increase stability score without increasing stability. For example, raising unfolded-state proteolysis resistance without raising the USM's predictions would yield a higher empirical resistance to proteolysis and thus a higher stability score, without actually increasing the stability of the protein.

To reduce the chance of this happening, we constructed new USMs (one for each of the two proteases used). These were constructed like the originals [26], with three differences. First, rather than learning one kernel, each model learned 100. Second, rather than taking the sum of the convolution vector of the kernel with the encoded protein sequence, each kernel's maximum activation over the entire convolution vector was taken, and these maxima were combined via a learned linear function. Finally, in addition to scrambles from Library 1, we used scrambles from Library 2.

### Reconciling multiple protein libraries

In addition to the protein designs described in [26, 32], our training corpus included designs from five other sets of proteins built using Rosetta or dTERMen. Because these were run with different batches of reagents, in different labs, at different times, by different people, some experimental variance likely affected the data. To compensate for this and calibrate results, we used a set of "ladder" proteins—present in all libraries—that spanned the range of $EC_{50}$ values. For each chip and each protease, we performed an orthogonal regression between observed $EC_{50}$ values of ladder proteins on the new chip and observed $EC_{50}$ values of ladder proteins on the original four chips. The resulting linear function was then applied to map the new $EC_{50}$ values into the same space as the original four libraries.

A second complication was introduced and mitigated in the chips of 100K designs developed and assayed for this manuscript. The proteins in the training set were embedded in the expression vector described in [26], which set a soft upper limit on the $EC_{50}$ values that could practically be assayed: beyond a certain value, the expression vector itself would start to suffer

measurable cleavage. For Corpora B and C, an updated version of the expression vector was used. This allows for a wider range of $EC_{50}$ values and hence stability scores, which will be valuable for future research. However, the EM and the Unfolded-State Model were trained with stability scores generated using the original expression vector, making numerical comparisons between new data and the models' predictions more difficult. To address this, we learned functions that mapped $EC_{50}$ values from the new expression vector to what we would have expected had we run the experiments with the original expression vector. We ran a repeat of the Round 4 chip from [26] with the new expression vector. We compared the resulting $EC_{50}$ values with those of the original Round 4 chip, and subjected them to a piecewise-linear orthogonal regression. Data up to a best-fit inflection point were constrained to lie along $y = x$; beyond the inflection point, slope was allowed to vary. This yielded a two-parameter model for each protease. After inverting these functions to map our new data into the original expression vector's space, we applied the orthogonal regression (described above) for reconciling multiple libraries. Data and best-fit functions are shown in S38 and S39 Figs.

## Evaluator model

The EM is built around a convolutional neural network. Its input is a fixed one-hot encoding layer for an amino acid sequence, with dimension $23 \times 175$. There are 20 amino acids permissible, along with codes indicating no amino acid ("X", i.e. padding), the start of a protein ("J"), and the end of a protein ("O"). A given sequence is encoded as a sparse matrix, mostly 0 but with a 1 corresponding to the identity of the amino acid at each location. During training and testing, it is situated at a random (but consistent for a given sequence) site within the 145 middle locations of the input layer, preceded by a run of "X" and then a "J", and followed by an "O" and then another run of "X". The random location shift prevents the model from mapping particular sites in a protein sequence to particular inputs, encouraging the learning of relational attributes.

The first convolutional layer comprises 400 kernels, each of them $23 \times 5$. Because only valid convolutions (i.e. those without edge effects) are used, the dimensionality of this layer is $171 \times 400$. The second and third convolutional layers contain, respectively, 200 $1 \times 9$ and 100 $1 \times 17$ kernels. All three layers use ReLU activation functions and 20% dropout during training.

The output of the last convolutional layer passes into two dense layers. One, for the secondary structure prediction pathway, is $6 \times 175$ units (one-hot encoding of loops, helices, strands, and "X"/"J"/"O" null/start/stop codes as in the input). This layer uses a softmax activation and represents the model's prediction of secondary structure. The other dense layer receiving input from the final convolutional layer consists of 80 ReLU-activated units. This layer passes in to a densely connected layer of 40 ReLU-activated units, and ultimately a 2-unit dense layer with a linear activation function—stability scores for chymotrypsin and trypsin. A third unit, whose value is the minimum of the two stability score outputs, is also appended.

There are two ways in which the EM departs from a standard CNN. First, in addition to predicting stability scores, the EM simultaneously predicts secondary structure sequences. Multi-task models have been found to improve performance in other domains [68]; we observed a small improvement in stability score prediction performance after adding this second output pathway. The secondary structure output was highly predictive of actual secondary structure labels, which were generated by applying DSSP [46] to tertiary structure models built by Rosetta during the design process. Levenshtein edit distance for secondary structure sequences averaged 2.7 characters per protein over the test set.

Second, the EM is trained using natural protein sequences in addition to stability data from designed proteins. We reasoned that a naturally occurring protein is more likely to be stable than a scrambled sequence composed of the same amino acids. During training, triplets of proteins are passed in: a designed protein, labeled with its experimental stability scores and secondary sequence; a natural protein sequence, with no associated label; and a scrambled version of the same natural protein sequence (again with no associated label). For designed proteins, the model's loss function encouraged learning how to map primary sequences to secondary sequences and stability scores. For natural proteins, the loss function imposed a penalty proportional to the predicted stability of the scramble minus that of the natural protein, i.e. encouraging the model to learn that natural proteins are more stable than scrambles. Inclusion of these natural proteins yielded another modest improvement in stability score prediction performance. We included 22,354 natural proteins whose length was between 25 and 65 amino acids from the UniProt database [24].

The loss function for the stability score outputs is

$$\mathcal{L}_s(X, \hat{X}) = \sqrt{\overline{(X - \hat{X})^2}} - 5\left(1 - \frac{\overline{(x_m - \hat{x}_m)^2}}{\overline{(x_m - \overline{x}_m)^2}}\right) - \left(1 - \frac{\overline{(x_{c,t} - \hat{x}_{c,t})^2}}{\overline{(x_{c,t} - \overline{x}_{c,t})^2}}\right).$$

Here, $X = (x_c, x_t, x_m)$ is the experimentally observed stability score vector corresponding to chymotrypsin, trypsin, and their minimum; $\hat{}$ indicates the predicted value; $\bar{}$ indicates the mean over a minibatch of the underlying quantity. This loss function encourages a low root-mean-squared error between predicted and actual stability scores (the first of the three terms). It also encourages a high coefficient of determination between the predicted and actual stability score (the second term) and, less strongly, between the predicted and actual individual protease stability scores (the third term).

The loss function for the comparator of natural and scrambled natural protein sequences is

$$\mathcal{L}_c(\hat{Y}, \tilde{Y}) = \frac{2.5}{1 + e^{\tilde{y}_m - \hat{y}_m}} + 0.001\sqrt{\overline{(\hat{y}_c - \hat{y}_t)^2}},$$

where $\hat{Y}$ and $\tilde{Y}$ are the model's predicted stability score vectors for the natural and scrambled natural proteins, respectively. This loss function encourages natural protein sequences to have higher predicted stability than scrambled natural protein sequences, though it places a cap on the bonus/penalty possible due to this difference. There is also mild encouragement for the two predicted protease stability scores for natural proteins to be similar to each other.

The loss function for the secondary structure output depends upon a bank of convolutional kernels that encode the rules by which secondary sequences can change from one position to the next. Each kernel corresponds to one of the six symbols in the library, {"L", "E", "H", "X", "J", "O"}, as indicated by the position of the 1 in the first column. The second column indicates with a -1 that the corresponding symbol is permitted to follow the symbol in question. For example, an amino acid in a loop ("L") may be followed by another loop amino acid or by the beginning of a strand ("E") or helix ("H"), or it may be the end of the protein (followed by "O"). It may not be followed by an "X" (because the end of a protein must be set off from empty space by an "O") or a "J" (because an amino acid that is part of a loop cannot appear before the beginning of the protein). The sum of the convolutions of these filters with the one-hot encoding of a protein's predicted secondary structure gives the number of times these rules are violated. Importantly for training, the convolution of these filters with the softmax activation output yields a measure of error with a gradient that can be followed to better

secondary structure predictions.

$$K = \left[ \begin{bmatrix} 1 & -1 \\ 0 & -1 \\ 0 & -1 \\ 0 & 0 \\ 0 & 0 \\ 0 & -1 \end{bmatrix}, \begin{bmatrix} 0 & -1 \\ 1 & -1 \\ 0 & 0 \\ 0 & 0 \\ 0 & 0 \\ 0 & 0 \end{bmatrix}, \begin{bmatrix} 0 & -1 \\ 0 & 0 \\ 1 & -1 \\ 0 & 0 \\ 0 & 0 \\ 0 & 0 \end{bmatrix}, \begin{bmatrix} 0 & 0 \\ 0 & 0 \\ 0 & 0 \\ 1 & -1 \\ 0 & -1 \\ 0 & 0 \end{bmatrix}, \begin{bmatrix} 0 & -1 \\ 0 & 0 \\ 0 & 0 \\ 0 & 0 \\ 1 & 0 \\ 0 & 0 \end{bmatrix}, \begin{bmatrix} 0 & 0 \\ 0 & 0 \\ 0 & 0 \\ 0 & -1 \\ 0 & 0 \\ 1 & 0 \end{bmatrix} \right]$$

We define this convolutional loss as follows:

$$\mathcal{L}_*(\hat{Z}) = \max_i \; clip(\hat{Z} * K_i),$$

where $\hat{Z}$ is the softmax activation of the secondary structure channel, *clip* forces a value to be between 0 and 1, and the maximum value of the convolution over all six kernels is evaluated at each position in the convolution vectors. If $\hat{Z}_{01}$ is the one-hot binarization of $\hat{Z}$, i.e. the matrix with a 1 corresponding to the highest of the six values in $\hat{Z}$ at each amino acid position and a 0 otherwise, we arrive at the final loss function for the secondary structure pathway:

$$\mathcal{L}_d(Z, \hat{Z}) = -\sum Z \, \log \, \hat{Z} + 0.8 \overline{\mathcal{L}_*(\hat{Z}_{01})} + 0.4 \, \max \, \mathcal{L}_*(\hat{Z}_{01}) + 0.1 \overline{\mathcal{L}_*(\hat{Z})} + 0.05 \, \max \, \mathcal{L}_*(\hat{Z}).$$

This is the categorical cross-entropy, plus four estimates of the invalidity of the predicted secondary sequence—two based on the one-hot binarization of the output and two on the raw softmax activation; two considering the mean invalidity over the whole sequence and two considering the worst position in the sequence.

The final loss function for the entire network, used in training, is then given by

$$\mathcal{L}(X, \hat{X}, \hat{Y}, \tilde{Y}, Z, \hat{Z}) = 0.2 \; \mathcal{L}_s(X, \hat{X}) + 0.1 \; \mathcal{L}_c(\hat{Y}, \tilde{Y}) + 2 \; \mathcal{L}_d(Z, \hat{Z}).$$

Coefficients on the different loss terms were selected because they encouraged rapid convergence during early tuning; empirically, the model's convergence is insensitive to modest changes of these coefficients. This loss was used with the Adadelta optimizer [69] to train the model, with each minibatch containing 64 designed proteins (and their stability scores and secondary structures), 64 natural proteins, and 64 disrupted natural proteins. Each designed protein was presented once per epoch, and training continued until validation loss (based on 10,000 validation samples randomly held-out from the training data set) failed to improve for five epochs. The saved state of the model five epochs previous was then recovered, i.e. the model with the best validation performance. This generally required 15–25 epochs of about 90 seconds each on an NVIDIA GeForce GTX 1080 Ti. The model was implemented using Keras 2.1.6 [70] with the TensorFlow 1.14.0 [71] backend.

## Generator model

The GM estimates a conditional distribution over primary sequences given a secondary sequence of the same length using a deep learning recurrent neural network encoder/decoder architecture with attention [45]. The model is conditioned on a secondary sequence which consists of "E", "H", and "L" tokens, corresponding to beta-sheets, alpha-helices and loops respectively. The output primary vocabulary is 20 amino acids plus a special stop token to

signify the end of a sequence. The model first encodes the secondary sequence into a sequence of compact vector space representation. Then the network decodes a primary sequence conditioned on the internal representation.

Since the decoder iteratively generates amino acids from left to right, the encoder is augmented with an attention mechanism that estimates which parts of the input secondary sequence are most salient to the position during decoding. The encoder and attention networks both use convolutional neural networks while the decoder is a long short-term memory recurrent neural network. In total, the model has 128M parameters and has no explicit encoding of biophysical properties—it only observes pairs of aligned secondary and primary sequences.

When generating primary sequences with the GM, we use a beam search to produce five likely primary sequences from each secondary sequence. The model iteratively builds up primary sequences, one amino acid at a time. At each iteration (e.g. iteration $N$), we have five intermediate primary sequences produced so far (e.g. of length $N$). A distribution over all possible next amino acids (e.g. the $N + 1^{th}$ amino acid) is calculated for each of the five sequences, and the five likeliest new sequences (each of length $N + 1$) are used for the next iteration.

To estimate the parameters of the GM, we considered proteins in Corpus A. These designs were all assayed for stability and ranged in length from 28 to 65 amino acids (mean 55). For each design, we examined secondary structure generated by the DSSP algorithm from the Rosetta-designed pdb [46]. 203 unique secondary topologies were present in this data set, with 10 having 100 or more designs. Our aim is to produce stable proteins, but the formulation of our model does not take stability into account. To overcome this, we subset Corpus A so that designs were represented in proportion to their stability score. This training data set shifted the mean stability from 0.73 to 0.91, using 46,448 unique protein designs. We then fit the parameters of our model by minimizing the cross-entropy of the model's predicted primary sequence with the reference primary sequence using stochastic gradient descent, given the secondary sequence as input. We stopped training after 65 iterations, by which point loss appeared to have converged. Optimization took around 24 hours training on four GTX 1080 Ti GPUs in parallel [72].

## Individual protein analyses

**Protein expression.** Genes encoding the designed protein sequences were synthesized and cloned into modified pET-29b(+) *E. coli* plasmid expression vectors (GenScript, N-terminal 8× His-tagged followed by a TEV cleavage site). For all the designed proteins, the sequence of the N-terminal tag used is MSHHHHHHHHSENLYFQSGGG (unless otherwise noted), which is followed immediately by the sequence of the designed protein. Plasmids were then transformed into chemically competent *E. coli* Lemo21 cells (NEB). Protein expression was performed using the Studier autoinduction media supplemented with appropriate antibiotics, and grown overnight at 37˚C. The cells were harvested by spinning at 4,000xg for 10 min and then resuspended in lysis buffer (300 mM NaCl, 30 mM Tris-HCL, pH 8.0, with 0.25% CHAPS for cell assay samples) with DNAse and protease inhibitor tablets. The cells were lysed with a QSONICA SONICATORS sonicator for 4 minutes total (2 minutes on time, 10 sec on-10 sec off) with an amplitude of 80%. Then the soluble fraction was clarified by centrifugation at 24,000g for 30 min. The soluble fraction was purified by Immobilized Metal Affinity Chromatography (Qiagen) followed by FPLC size-exclusion chromatography (Superdex 75 10/300 GL, GE Healthcare). All protein samples were characterized with SDS-PAGE with the purity higher than 95%. Protein concentrations were determined by absorbance at 280 nm measured using a NanoDrop spectrophotometer (Thermo Scientific) using predicted extinction coefficients.

**Circular dichroism.** Far-ultraviolet CD measurements were carried out with an JASCO-1500 equipped with a temperature-controlled multi-cell holder. Wavelength scans were measured from 240 to 190 nm at 24°C. Temperature melts were recorded by monitoring CD signal at 222 nm in steps of 2°C per minute with 5s of equilibration time. Wavelength scans and temperature melts were performed using 0.3 mg/ml protein in PBS buffer (20mM NaPO4, 150mM NaCl, pH 7.4) with a 1 mm path-length cuvette. Raw CD mdeg data were converted to molar ellipticity using standard methods [73].

**Crystallography sample preparation, data collection, and analysis.** Crystal screening was performed using Mosquito LCP by STP Labtech. Crystals were grown in 10% PEG 20000, 20% PEG MME 550, 0.2M Sodium formate; 0.2M Ammonium acetate; 0.2M Sodium citrate tribasic dihydrate; 0.2M Potassium sodium tartrate tetrahydrate; 0.2M Sodium oxamate and 0.1M MES/imidazole pH 6.5. Crystals were subsequently harvested in a cryo-loop and flash frozen directly in liquid nitrogen for synchrotron data collection. Data was collected on 24-ID-C at NECAT, APS. X-ray intensities and data reduction were evaluated and integrated using XDS [74] and merged/scaled using Pointless/Aimless in the CCP4 program suite [75].

**Table 2. Crystallographic data collection and refinement statistics.**

| | nmt_0994_guided_02 (PDB: 7KUW) |
| --- | --- |
| **Data collection** | |
| Space group | $P4_32_12$ |
| Cell dimensions | |
| *a, b, c* (Å) | 52.26, 52.26, 47.35 |
| $\alpha, \beta, \gamma$ (°) | 90, 90, 90 |
| Resolution (Å) | 52.27–2.43 (2.53–2.43)[a] |
| No. of unique reflections | 2746 (299) |
| $R_{merge}$ | 0.471 (2.479) |
| $R_{pim}$ | 0.107 (0.579) |
| $I/\sigma(I)$ | 6.5 (1.1) |
| $CC_{1/2}$ | 0.987 (0.421) |
| Completeness (%) | 100.0 (100.0) |
| Redundancy | 22.6 (19.4) |
| **Refinement** | |
| Resolution (Å) | 36.96–2.43 (2.51– 2.43) |
| No. of reflections | 12717 (267) |
| $R_{work}$ / $R_{free}$ (%) | 24.7 / 28.1 (34.0 / 31.8) |
| No. atoms | 507 |
| Protein | 507 |
| Ion/Ligand | 0 |
| Water | 0 |
| Ramachandran Favored/allowed | 98.33/1.67 |
| Outlier (%) | 00.00 |
| r.m.s. deviations | |
| Bond lengths (Å) | 0.003 |
| Bond angles (°) | 0.420 |
| $B_{factors}$ (Å$^2$) | |
| Protein | 67.08 |

Data were collected from one single crystal.

[a]Values in parentheses are for the highest-resolution shell.

Starting phases were obtained by molecular replacement using Phaser [76] with Rosetta prediction as the search model. Sidechains were rebuilt and the model was refined with Rosetta-Phenix [77]. Manual rebuilding in Coot [78] and cycles of Phenix refinement were used to build the final model. The final model was evaluated using MolProbity [79]. Structure deposited to PDB (PDB ID **7KUW**). Data collection and refinement statistics are recorded in Table 2.

## Supporting information

**S1 Fig. Protein stability assay.** Designed protein sequences are fused in-frame with the yeast surface protein Aga2 and a C-terminal myc tag (for expressing on the surface of yeast, and tagging with FITC, respectively), encoded by an oligonucleotide library. The library is expressed on the surface of yeast cells (one design per cell). Titration with proteases, followed by FACS and deep sequencing, yields a measurement of each design's resistance to proteolysis. Separately, the USM predicts the intrinsic unfolded-state resistance of each design, and this value is subtracted from the observed resistance. The resulting stability score is a measure of the protection against proteolysis conferred by the tertiary structure of the protein.
(EPS)

**S2 Fig. New unfolded-state model.** (**A**) The new Unfolded State Model (USM) improved correlation coefficients between predicted and actual $EC_{50}$ values for both trypsin and chymotrypsin. (**B**) Root mean squared errors between predicted and actual $EC_{50}$ values also improved for the new USM relative to the original. (**C**) By chance, the set of designed proteins is likely to have a similar distribution of unfolded-state protease resistances to a set of scrambled versions of those designs. If the medians of the two predicted distributions differ, therefore, it suggests that the USM is biased. Here we see that the difference between median USM-predicted design $EC_{50}$s and and median USM-predicted scramble $EC_{50}$s is lower for the new USM, suggesting that it is less biased despite its greater complexity.
(EPS)

**S3 Fig. Relationship between $EC_{50}$ values for trypsin and chymotrypsin.**
(EPS)

**S4 Fig. Relationship between stability scores for trypsin and chymotrypsin.** The increased correlation coefficient versus that seen between $EC_{50}$ values for trypsin and chymotrypsin (S3 Fig) suggests that stability score (which corrects for differences in unfolded-state susceptibility by subtracting off the prediction of the USM) is a better measure of stability than uncorrected $EC_{50}$. The original USM described in [26] achieved an $r^2$ of 0.62 on the same data.
(EPS)

**S5 Fig. Summary of Corpus A mini-protein designs.**
(EPS)

**S6 Fig. Characteristics of Corpus A mini-protein designs.** Different panels show data for different Libraries (S5 Fig), or different groups of designs within the same Library (we describe these groups in more detail in Materials and Methods). Within each panel, we group designs by topology (y axis) according to the naming scheme described in Materials and Methods. The leftmost plot in each panel shows distributions of stability scores for designs (blue) and scrambles (orange), where boxes show the quartiles, whiskers extend to either the end of the distribution or a maximum of 1.5-fold of the inter-quartile range past the outer quartiles, and diamonds show outliers beyond this range. Successive plots show the fraction of "stable" proteins (stability score > 1), the distributions of sequence lengths, and the number of proteins in

each group. (**A**) Data from [26]. The very narrow length distributions reflect design protocols that specified one or two precise lengths for each topology. Scrambles longer than their designed counterparts retained amino-acid padding used to normalize the lengths of a subset of proteins to 50 amino acids. Although this padding was stripped from design sequences, it was retained in scrambled sequences when we trained the new USM in this study. (**B**) Protein designs from [32]; we assayed stability scores for these designs in this study. (**C**) Blueprint-based Rosetta designs. Narrow length distributions are again because of the design protocols used. (**D**) Designs created using dTERMen software. (**E**) Designs created using a TERM-based Rosetta protocol.
(EPS)

**S7 Fig. Generalization performance by protein class.** Performance for the EM on each held-out protein class, after training on all other classes.
(EPS)

**S8 Fig. Evaluator model predicts stability of perturbed sequences.** Observed versus predicted stability scores for proteins after being subjected to each of the fourteen sequence perturbations. In contrast to S9 Fig, where we showed stability score changes for stable (or predicted-stable) proteins, here we show stability scores for all proteins.
(EPS)

**S9 Fig. Evaluator model predicts destabilizing effects of perturbations.** Impact of fourteen types of sequence perturbation on predicted and observed stability. `Substitutions`, `Insertions`, and `Deletions` are single-site changes at random locations. Their respective `Central` versions are at random locations excluding the first and last ten amino acids. `Cycled` proteins had the indicated number of amino acids moved from the beginning of the sequence to the end. `Half` sequences preserved the indicated 50% of the sequence, discarding the rest. `Reversal` proteins were simple sequence reversals. Only cases where the predicted or actual stability score of the base protein was $\geq 1$ are considered, in order to illustrate the effects (and predicted effects) of each manipulation on proteins with some real or predicted baseline stability.
(EPS)

**S10 Fig. Refinement of stable designs far from EC$_{50}$ ceiling.** Effects on stability scores of guided and random substitutions within expert-designed proteins with base-design stability score $< 0.7$ and EC$_{50}$ values $< 4$ (before vector reconciliation) for both proteases.
(EPS)

**S11 Fig. Trypsin stability scores with refinement.** Effects on trypsin stability scores of guided and random substitutions within expert-designed proteins.
(EPS)

**S12 Fig. Chymotrypsin stability scores with refinement.** Effects on chymotrypsin stability scores of guided and random substitutions within expert-designed proteins.
(EPS)

**S13 Fig. Trypsin EC$_{50}$ values with refinement.** Effects on trypsin EC$_{50}$ values of guided and random substitutions within expert-designed proteins.
(EPS)

**S14 Fig. Chymotrypsin EC$_{50}$ values with refinement.** Effects on chymotrypsin EC$_{50}$ values of guided and random substitutions within expert-designed proteins.
(EPS)

**S15 Fig. Refined trypsin EC$_{50}$ values for stable designs far from EC$_{50}$ ceiling.** Effects on trypsin EC$_{50}$ values of guided and random substitutions within expert-designed proteins, restricted to proteins with base-design stability score $> 0.7$ and EC$_{50}$ values $< 4$ (before vector reconciliation).
(EPS)

**S16 Fig. Refined chymotrypsin EC$_{50}$ values for stable designs far from EC$_{50}$ ceiling.** Effects on chymotrypsin EC$_{50}$ values of guided and random substitutions within expert-designed proteins, restricted to proteins with base-design stability score $> 0.7$ and EC$_{50}$ values $< 4$ (before vector reconciliation).
(EPS)

**S17 Fig. Raw scatter plots of experimental stability versus percent identity.** Experimental stability for GM proteins as novelty increases. Novelty is determined by the maximum percent identity as computed by BLAST over all designs in the training data set (i.e. the percent identity to the most similar training design).
(EPS)

**S18 Fig. Trypsin stability scores with refinement.** Effects of guided and random substitutions within GM designs on trypsin stability scores.
(EPS)

**S19 Fig. Chymotrypsin stability scores with refinement.** Effects of guided and random mutations of GM designs on chymotrypsin stability scores.
(EPS)

**S20 Fig. Trypsin EC$_{50}$ values with refinement.** Effects of guided and random mutations of GM designs on trypsin EC$_{50}$ values.
(EPS)

**S21 Fig. Chymotrypsin EC$_{50}$ values with refinement.** Effects of guided and random mutations of GM designs on chymotrypsin EC$_{50}$ values.
(EPS)

**S22 Fig. Effects and frequencies of guided single-site substitutions on protein stability.** For each original amino acid (indexed on the Y-axis) and each changed-to amino acid (indexed on the X-axis), the mean effect on stability when that substitution was guided by the EM is computed and plotted for each from-to pair that was represented in the data; redder circles indicate that guided substitutions were on average stabilizing, bluer circles indicate that they were on average destabilizing. Circles with heavy black outlines showed a significant deviation from zero (two-tailed t-test, $p < 0.05$ uncorrected). Bar graphs indicate mean differences over all changed-to amino acids for each original amino acid (left) and all original amino acids for each changed-to amino acid (bottom). Circle area beyond a minimal baseline size indicates the number of times the corresponding substitution was made. Numbers on the axes beside/below the amino acid abbreviations indicate the total number of times the corresponding amino acid was mutated from/to.
(EPS)

**S23 Fig. Effects and frequencies of random single-site substitutions on protein stability.** Conventions are the same as in S22 Fig.
(EPS)

**S24 Fig. Circular dichroism.** Thermal stability of the designed proteins monitored by circular dichroism spectroscopy (CD). Left, full-spectral scans at 24˚C and 84˚C for each variant between 200–240nm wavelength. Right, MRE at 222 nm as a function of temperature.
(EPS)

**S25 Fig. Structural predictions.** Ab initio Rosetta predictions of the structures of the twelve designs subjected to additional testing. The top row shows the four designs which were refined to yield the designs in the middle row; the bottom row shows the four designs for which only refined versions were evaluated. Note that nmt_0994_guided_02 was evaluated crystallographically and found to have a somewhat different structure from that predicted here (Fig 7B).
(EPS)

**S26 Fig. Stability scores with refinement and no USM decreases.** Effects of guided and random substitutions within expert-designed proteins, when guided substitutions are not permitted to decrease USM $EC_{50}$ predictions.
(EPS)

**S27 Fig. Stability scores with refinement and no USM decreases.** Effects of guided and random substitutions within GM designs with a maximum sequence identity to any expert-designed protein of less than 50%, when guided substitutions are not permitted to decrease USM $EC_{50}$ predictions.
(EPS)

**S28 Fig. Effects and frequencies of guided single-site substitutions when guided mutations are not permitted to reduce USM-predicted unfolded-state resistance.** This restriction reduces the opportunities for the refinement process to improve stability score simply by exploiting flaws in the USM. Conventions are the same as in S22 Fig.
(EPS)

**S29 Fig. Effects and frequencies of guided single-site substitutions when guided mutations are not permitted to reduce USM-predicted unfolded-state resistance or to change amino acids to tryptophan.** Conventions are the same as in S22 Fig.
(EPS)

**S30 Fig. Stability scores with refinement, no USM decreases, no W.** Effects of guided substitutions within expert-designed proteins, when they are not permitted to decrease USM $EC_{50}$ predictions or to change amino acids to tryptophan.
(EPS)

**S31 Fig. Stability scores with refinement, no USM decreases, no W.** Effects of guided and random substitutions within GM designs with a maximum sequence identity to any expert-designed protein of less than 50%, when guided substitutions are not permitted to decrease USM $EC_{50}$ predictions and neither guided nor random substitutions are allowed to change amino acids to tryptophan.
(EPS)

**S32 Fig. Stability scores with refinement, no USM decreases, no AFILMVWY.** Effects of guided and random substitutions within GM designs with a maximum sequence identity to any expert-designed protein of less than 50%, when guided substitutions are not permitted to decrease USM $EC_{50}$ predictions and neither guided nor random substitutions are allowed to change amino acids to alanine, phenylalanine, isoleucine, leucine, methionine, valine, tryptophan, or tyrosine.
(EPS)

**S33 Fig. Stability scores with refinement, no USM decreases, no AFILMVWY.** Effects of guided substitutions within expert-designed proteins, when they are not permitted to decrease USM $EC_{50}$ predictions or to change amino acids to alanine, phenylalanine, isoleucine, leucine, methionine, valine, tryptophan, or tyrosine.
(EPS)

**S34 Fig. Effects and frequencies of guided single-site substitutions when guided mutations are not permitted to reduce USM-predicted unfolded-state resistance or to change amino acids to any of the following: Alanine, phenylalanine, isoleucine, leucine, methionine, valine, tryptophan, or tyrosine.** Conventions are the same as in S22 Fig.
(EPS)

**S35 Fig. Refinement of scrambles, with no USM decreases.** Effects of guided substitutions within scrambled expert-designed proteins, when they are not permitted to decrease USM $EC_{50}$ predictions.
(EPS)

**S36 Fig. Refinement of scrambles, with no USM decreases, no W.** Effects of guided substitutions within scrambled expert-designed proteins, when they are not permitted to decrease USM $EC_{50}$ predictions or to change amino acids to tryptophan.
(EPS)

**S37 Fig. Refinement of scrambles, with no USM decreases, no AFILMVWY.** Effects of guided substitutions within scrambled expert-designed proteins, when they are not permitted to decrease USM $EC_{50}$ predictions or to change amino acids to alanine, phenylalanine, isoleucine, leucine, methionine, valine, tryptophan, or tyrosine.
(EPS)

**S38 Fig. Mapping chymotrypsin $EC_{50}$ values between expression vectors.** Chymotrypsin $EC_{50}$ values observed for proteins embedded in the second-generation expression vector (Y-axis) plotted against $EC_{50}$ values observed for the same proteins embedded in the original expression vector. The heavy red line shows the function learned through orthogonal regression to map the first-generation values to the second-generation values. The function is constrained to be $y = x$ below a parametrized inflection point, at which point its slope is allowed to range to best fit the orthogonal regression. Optimal parameters for this fit are shown. By inverting this function, we can map chymotrypsin $EC_{50}$ values from those observed using the second-generation expression vector back to what they would have been had they been tested with the first-generation vector.
(EPS)

**S39 Fig. Mapping trypsin $EC_{50}$ values between expression vectors.** Trypsin $EC_{50}$ values observed for proteins embedded in the second-generation expression vector (Y-axis) plotted against $EC_{50}$ values observed for the same proteins embedded in the original expression vector. Conventions are as in S38 Fig.
(EPS)

## Acknowledgments

We thank Tobin Sosnick for constructive criticism and thoughtful suggestions about the manuscript. We thank the Texas Advanced Computing Center for computational resources and support. We thank Ivan Anishchenko for his assistance in running trRosetta. We thank Michelle DeWitt and Ayesha Saleem at the Institute for Protein Design Core Labs for

assistance in protein purification. We thank the staff at Northeastern Collaborative Access Team at Advanced Photon Source for the beamtime.

## Author Contributions

**Conceptualization:** Jedediah M. Singer, Scott Novotney, Devin Strickland, Hugh K. Haddox, Nicholas Leiby, David Baker, Eric Klavins.

**Data curation:** Jedediah M. Singer, Scott Novotney, Devin Strickland, Hugh K. Haddox, Nicholas Leiby, Asim K. Bera, Francis C. Motta, Longxing Cao, Eva-Maria Strauch, Tamuka M. Chidyausiku, Alex Ford, Alexander Zaitzeff, Craig O. Mackenzie, Hamed Eramian, Gevorg Grigoryan.

**Formal analysis:** Jedediah M. Singer, Scott Novotney, Hugh K. Haddox, Nicholas Leiby, Asim K. Bera, Alexander Zaitzeff, Frank DiMaio, Lance J. Stewart.

**Funding acquisition:** Jedediah M. Singer, Scott Novotney, Devin Strickland, Francis C. Motta, Matthew Vaughn, Lance J. Stewart, David Baker, Eric Klavins.

**Investigation:** Devin Strickland, Hugh K. Haddox, Cameron M. Chow, Anindya Roy, Asim K. Bera, Lance J. Stewart.

**Methodology:** Jedediah M. Singer, Scott Novotney, Devin Strickland, Hugh K. Haddox, Nicholas Leiby, Asim K. Bera, Francis C. Motta, Ethan Ho, Craig O. Mackenzie, Gevorg Grigoryan.

**Project administration:** Jedediah M. Singer, Scott Novotney, Cameron M. Chow, Anindya Roy, Lance J. Stewart.

**Resources:** Longxing Cao, Eva-Maria Strauch, Tamuka M. Chidyausiku, Alex Ford, Lance J. Stewart.

**Software:** Jedediah M. Singer, Scott Novotney, Devin Strickland, Hugh K. Haddox, Nicholas Leiby, Francis C. Motta, Ethan Ho, Craig O. Mackenzie, Gevorg Grigoryan.

**Supervision:** Jedediah M. Singer, Scott Novotney, Devin Strickland, Gabriel J. Rocklin, Matthew Vaughn, Lance J. Stewart, David Baker, Eric Klavins.

**Validation:** Jedediah M. Singer, Hugh K. Haddox, Ethan Ho.

**Visualization:** Jedediah M. Singer, Scott Novotney, Hugh K. Haddox, Nicholas Leiby, Cameron M. Chow, Anindya Roy.

**Writing – original draft:** Jedediah M. Singer, Scott Novotney, Devin Strickland, Hugh K. Haddox.

**Writing – review & editing:** Jedediah M. Singer, Scott Novotney, Devin Strickland, Hugh K. Haddox, Nicholas Leiby, Gabriel J. Rocklin, Cameron M. Chow, Anindya Roy, Asim K. Bera, Francis C. Motta, Alexander Zaitzeff, Lance J. Stewart.

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
