## [Decision Letter · Decision Letter 0]

4 Jan 2022

PONE-D-21-36596Large-scale design and refinement of stable proteins using sequence-only modelsPLOS ONE

Dear Dr. Singer,

Thank you for submitting your manuscript to PLOS ONE. After careful consideration, we feel that it has merit but does not fully meet PLOS ONE’s publication criteria as it currently stands. Therefore, we invite you to submit a revised version of the manuscript that addresses the points raised during the review process.

As all three reviewers praised the quality of the work presented in this manuscript, they only suggested minor corrections, especially the interpretation of the R2 values obtained  with the sequence-only model.

We look forward to receiving your revised manuscript.

Kind regards,

Jean-Christophe Nebel, Ph.D

Academic Editor

PLOS ONE

Journal Requirements:

JMS and AZ are employed by Two Six Technologies, which has filed a patent on a portion of the technology described in this manuscript.

4. We noted in your submission details that a portion of your manuscript may have been presented or published elsewhere. [Some of the amino acid sequences (Corpus A, Libraries 1 and 2) and some of the stability scores (Corpus A, Library 1) have been previously published, as detailed in Figure S5.] Please clarify whether this publication was peer-reviewed and formally published. If this work was previously peer-reviewed and published, in the cover letter please provide the reason that this work does not constitute dual publication and should be included in the current manuscript.

Reviewers' comments:

Reviewer's Responses to Questions

**Comments to the Author**

1. Is the manuscript technically sound, and do the data support the conclusions?

Reviewer #1: Yes

Reviewer #2: Yes

Reviewer #3: Yes

2. Has the statistical analysis been performed appropriately and rigorously? 

Reviewer #1: Yes

Reviewer #2: Yes

Reviewer #3: Yes

3. Have the authors made all data underlying the findings in their manuscript fully available?

Reviewer #1: Yes

Reviewer #2: Yes

Reviewer #3: Yes

4. Is the manuscript presented in an intelligible fashion and written in standard English?

Reviewer #1: Yes

Reviewer #2: Yes

Reviewer #3: Yes

5. Review Comments to the Author

Reviewer #1: This reviewer very much enjoyed reading the manuscript by Jedediah, Singer, Baker, et al., which is an interesting work regarding the prediction of protein stability based only on its amino acid sequence. The authors used a home-made convolutional neural network model. Accordingly, they combined a previously constructed parallel oligo library synthesis, yeast surface display and next-generation sequencing to generate very large datasets. In order to evaluate the performance of this model, 5 out of the 8 monomeric/oligomeric designs were predicted to be thermostable, and 4 do not unfold even at a temperature of 99°C. The authors succeeded in their challenging endeavor, thereby providing a new methodology to predicting protein stability using only sequences as input. Apparently, the authors have made acceptable responses in an earlier submission. I suggest that this manuscript can be suitable for Plos One after addressing the following concerns:

1. I noticed that the regression R2 (goodness-of-fit) values are pretty low (e.g. 0.38, 0.48, etc) when predicting stability with a sequence-only model. Is it reasonable? Can the authors provide some references or experimental details to demonstrate that these R2 values are actually acceptable?

2. The authors employed trRosetta to predict the structure that they obtained from their designs, and found that it displays good accuracy with the solved X-ray structure (ca. RMSD of 3.7 A). However, it would be even more interesting if the authors were to use RoseTTAFold or Alphafold2 for the predictions and provide a comparison of the accuracy. I do not expect that this will take too much time.

3. Line 117, add “as” before the Evaluator Model.

4. Since this article can be viewed as another advance in merging rational design and directed enzyme evolution, the recent review of methodology development by G. Qu, et al could be added (Angew. Chem. Int. Ed. 2020, 59, 13204-13231).

Reviewer #2: The manuscript presents a comprehensive study that combines interestingly in vitro and in silico experiments as an attempt to primarily develop a data-driven paradigm to assess the stability of small proteins. This article comprises two contributions the one related to the prediction of stability and the other one that is related to generating primary sequences of novel stable proteins from secondary sequences. In my turn, I would like to thank the three reviewers from PLoS CB for their thorough reviews; authors were able – to some extent - to address/answer their main concerns. In short, the manuscript does fulfill the requirements of this journal and I would recommend it.

I still have some minor concerns authors may take into consideration:

The abstract does not reflect the whole picture. In vitro experiments that constitute the inputs to the machine learning models are not mentioned at all. In the same context, once the information aforementioned is included, authors also may want to mention how stability is measured. I believe a couple of additional sentences – as long as it does not exceed the limit of words – would make the abstract – to some context - more comprehensive and more reflective to the extensive study readers are about to explore.

The only piece of information that describes a ‘supposed short literature review’ on in vitro stability experiments is found in lines 54-55: “This presents a particular problem for modeling protein stability because, historically, experimental measurements of stability have been laborious” without even any reference.

In lines 125-128 authors first used r2 (Person correlation coefficient) then introduced R2 (coefficient of determination) by relying on a fact “R2 is more stringent than r2 because it does not assume a linear regression of model predictions onto data and can be negative when the model performs worse than simply predicting the sample mean.”. I simply may not agree with that! the ‘default’ metric to be used when dealing with the question: “How well the predicted values match?” is definitely R2. Consequently, this is the first choice in this regard that should be mentioned without even explaining the reason behind using it. However, Pearson’s is ‘almost’ useless when quantifying a model performance. Accordingly, authors are asked to explain the added value of involving Pearson’s they are expecting. In the same context, I have same comments in Figure 2 (where r2 is mentioned and not R2). What about table I? can both values (experimentally measured melting temperatures And antilogs of stability scores) be compared value-to-value, i.e. the same range as in figure 2? If the answer is positive them R2 is used, otherwise the Pearson’s is correct, however, a short explanation is to be added in this regard. In the same context, authors described the results of Table I as “modest correlation”. With such values of p-value isn’t “modest” a bit an exaggerated description?

In lines 300-301, authors explain why cysteine was excluded from the library by stating that “stability assay is incompatible with disulfide bonds”. I see that such a crucial point in the whole study deserves more elaboration.

Reviewer #3: The manuscript presents a machine learning system supporting prediction of a protein stability, which could be used for evaluation of the stability upon introduced mutations or for designing a synthetic stable protein. Within the manuscript, the authors included comments from previously invited 3 referees and the rebuttal. This is a very unusual, though a very recommendable practice, which allows following the history of the manuscript and avoiding raising the same issues.

The manuscript is well written, the study presented in detail. Not only did it present the computational model but also evaluated it with newly expressed proteins. Although I agree with some of the criticisms raised by previous reviewers, but I understand that the authors are not able to fix all the issues within this concept. In general, I think the manuscript is worth publishing in PLOS One.

I would, however, raise a few other issues which bothered me while reading:

1) I have not found any information on availability of the software

2) References seem a bit outdated, especially I have not found any discussion with regard to more general methods for modeling proteins. Although I understand that they have a different goal but (see below)

3) With regard to the release of very accurate tool Alpha Fold 2, I am wondering if such general tools could not be very useful for predicting protein stability. For example, based on the deviation of modeled structure and general (secondary structure) assumptions for the modeled protein. Please discuss.

6. PLOS authors have the option to publish the peer review history of their article (what does this mean?). If published, this will include your full peer review and any attached files.

Reviewer #1: No

Reviewer #2: No

Reviewer #3: No

---

## [Author Response · Author response to Decision Letter 0]

25 Jan 2022

We thank the editor and the reviewers for their comments, and for the opportunity to address them. We included with our resubmission a Response to Reviewers, which has our responses inline in bold. I am pasting the content of that document into this text input field, as well; it does not accept formatting, so our responses are not in bold.

Comments from editor:

1. Please ensure that your manuscript meets PLOS ONE's style requirements, including those for file naming. The PLOS ONE style templates can be found at  https://journals.plos.org/plosone/s/file?id=wjVg/PLOSOne_formatting_sample_main_body.pdf and  https://journals.plos.org/plosone/s/file?id=ba62/PLOSOne_formatting_sample_title_authors_affiliations.pdf

We believe we have followed all style requirements. We are using the PLOS LaTeX template, which should ensure this as long as we have used it correctly, but we also looked over the links provided here and saw nothing amiss. If, despite this verification, there is something wrong that we have missed, please let us know and we will fix it. Thank you!

We are unsure what information is in the “Funding Information” section, or where this section is. The statement in the “Financial Disclosure” is correct—please use it. To be extra-careful, we have reproduced it in the cover letter. Thank you!

3. Thank you for stating the following in the Competing Interests section:  JMS and AZ are employed by Two Six Technologies, which has filed a patent on a portion of the technology described in this manuscript.

This does not alter our adherence to PLOS ONE policies on sharing data and materials. We have included the updated statement in the cover letter.

4. We noted in your submission details that a portion of your manuscript may have been presented or published elsewhere. [Some of the amino acid sequences (Corpus A, Libraries 1 and 2) and some of the stability scores (Corpus A, Library 1) have been previously published, as detailed in Figure S5.] Please clarify whether this publication was peer-reviewed and formally published. If this work was previously peer-reviewed and published, in the cover letter please provide the reason that this work does not constitute dual publication and should be included in the current manuscript.

Some work has been previously published in peer-reviewed manuscripts, as noted. The aim of our work, however, is not to publish (or re-publish) these amino acid sequences or stability scores; rather, we are using them to train and evaluate stability prediction models. We do include them with the other data that we are publishing for convenience. We have clarified this in the cover letter.

5. Please review your reference list to ensure that it is complete and correct. If you have cited papers that have been retracted, please include the rationale for doing so in the manuscript text, or remove these references and replace them with relevant current references. Any changes to the reference list should be mentioned in the rebuttal letter that accompanies your revised manuscript. If you need to cite a retracted article, indicate the article’s retracted status in the References list and also include a citation and full reference for the retraction notice. 

We searched for each of our references in retractiondatabase.org, by DOI when available and by title otherwise, and found no hits. We are not aware of having cited any retracted articles.

Comments from reviewers:

Reviewer #1: This reviewer very much enjoyed reading the manuscript by Jedediah, Singer, Baker, et al., which is an interesting work regarding the prediction of protein stability based only on its amino acid sequence. The authors used a home-made convolutional neural network model. Accordingly, they combined a previously constructed parallel oligo library synthesis, yeast surface display and next-generation sequencing to generate very large datasets. In order to evaluate the performance of this model, 5 out of the 8 monomeric/oligomeric designs were predicted to be thermostable, and 4 do not unfold even at a temperature of 99°C. The authors succeeded in their challenging endeavor, thereby providing a new methodology to predicting protein stability using only sequences as input. Apparently, the authors have made acceptable responses in an earlier submission. I suggest that this manuscript can be suitable for Plos One after addressing the following concerns:  1. I noticed that the regression R2 (goodness-of-fit) values are pretty low (e.g. 0.38, 0.48, etc) when predicting stability with a sequence-only model. Is it reasonable? Can the authors provide some references or experimental details to demonstrate that these R2 values are actually acceptable?

We discuss this in the paragraph surrounding Line 130. Trypsin and chymotrypsin stability scores should ideally be equal, assuming no noise in the experimental assay and a perfect unfolded state model. However, the R2 (and r2) values between the per-protease stability scores are close to 0.4, suggesting a rather low limit due to experimental noise and flaws in the unfolded state model. One of the points that we hope comes through in this paper is that (as we say in line 230) even an imperfect model trained on noisy data can guide large improvements to stability.

 2. The authors employed trRosetta to predict the structure that they obtained from their designs, and found that it displays good accuracy with the solved X-ray structure (ca. RMSD of 3.7 A). However, it would be even more interesting if the authors were to use RoseTTAFold or Alphafold2 for the predictions and provide a comparison of the accuracy. I do not expect that this will take too much time.

We replaced the trRosetta prediction with an AlphaFold2 prediction, which slightly improved CA RMSD (to 3.4 A).

 3. Line 117, add “as” before the Evaluator Model.

Added “which we call”, which we think is even clearer.

 4. Since this article can be viewed as another advance in merging rational design and directed enzyme evolution, the recent review of methodology development by G. Qu, et al could be added (Angew. Chem. Int. Ed. 2020, 59, 13204-13231).

Thank you for highlighting this paper. We agree that it is both interesting and relevant, and have added it to the discussion.

Reviewer #2: The manuscript presents a comprehensive study that combines interestingly in vitro and in silico experiments as an attempt to primarily develop a data-driven paradigm to assess the stability of small proteins. This article comprises two contributions the one related to the prediction of stability and the other one that is related to generating primary sequences of novel stable proteins from secondary sequences. In my turn, I would like to thank the three reviewers from PLoS CB for their thorough reviews; authors were able – to some extent - to address/answer their main concerns. In short, the manuscript does fulfill the requirements of this journal and I would recommend it.  I still have some minor concerns authors may take into consideration:  The abstract does not reflect the whole picture. In vitro experiments that constitute the inputs to the machine learning models are not mentioned at all. In the same context, once the information aforementioned is included, authors also may want to mention how stability is measured. I believe a couple of additional sentences – as long as it does not exceed the limit of words – would make the abstract – to some context - more comprehensive and more reflective to the extensive study readers are about to explore. 

Well taken, thank you. We have updated the abstract and believe it better represents the manuscript.

The only piece of information that describes a ‘supposed short literature review’ on in vitro stability experiments is found in lines 54-55: “This presents a particular problem for modeling protein stability because, historically, experimental measurements of stability have been laborious” without even any reference. 

We have added citations that demonstrate how labor-intensive it is to gather stability measurements using traditional techniques (e.g. ProThermDB has ~31.5k proteins’ thermal stability, after collecting many years of work by many scientists). We hope the interested reader can use these citations as a starting point for further exploration, but feel that more extensive discussion of this history would stretch the already expansive bounds of this manuscript’s scope.

 In lines 125-128 authors first used r2 (Person correlation coefficient) then introduced R2 (coefficient of determination) by relying on a fact “R2 is more stringent than r2 because it does not assume a linear regression of model predictions onto data and can be negative when the model performs worse than simply predicting the sample mean.”. I simply may not agree with that! the ‘default’ metric to be used when dealing with the question: “How well the predicted values match?” is definitely R2. Consequently, this is the first choice in this regard that should be mentioned without even explaining the reason behind using it. However, Pearson’s is ‘almost’ useless when quantifying a model performance. Accordingly, authors are asked to explain the added value of involving Pearson’s they are expecting. In the same context, I have same comments in Figure 2 (where r2 is mentioned and not R2). 

We agree that R2 is the right metric here. We do include r2 as well, in some places, because some readers of earlier versions have asked for it; discussion of the two is intended to satisfy their requests while encouraging them to consider the better metric. We agree that figure 2 should report R2, and thank the reviewer for flagging this. We have made the change.

What about table I? can both values (experimentally measured melting temperatures And antilogs of stability scores) be compared value-to-value, i.e. the same range as in figure 2? If the answer is positive them R2 is used, otherwise the Pearson’s is correct, however, a short explanation is to be added in this regard. In the same context, authors described the results of Table I as “modest correlation”. With such values of p-value isn’t “modest” a bit an exaggerated description? 

For table 1, the model is making predictions in units of stability score, and we are comparing those to melting temperatures; while taking the antilog of the stability score should enable a linear relationship between the two, we would have to learn both the slope and the intercept of that relationship in order to properly relate the different units (which is what Pearson’s r does). We have added a brief explanation. We have no argument with your objection about “modest”, and have weakened the language there accordingly.

 In lines 300-301, authors explain why cysteine was excluded from the library by stating that “stability assay is incompatible with disulfide bonds”. I see that such a crucial point in the whole study deserves more elaboration.

We have expanded the explanation and added a citation to direct the reader interested in more detail to the original study.

Reviewer #3: The manuscript presents a machine learning system supporting prediction of a protein stability, which could be used for evaluation of the stability upon introduced mutations or for designing a synthetic stable protein. Within the manuscript, the authors included comments from previously invited 3 referees and the rebuttal. This is a very unusual, though a very recommendable practice, which allows following the history of the manuscript and avoiding raising the same issues.  The manuscript is well written, the study presented in detail. Not only did it present the computational model but also evaluated it with newly expressed proteins. Although I agree with some of the criticisms raised by previous reviewers, but I understand that the authors are not able to fix all the issues within this concept. In general, I think the manuscript is worth publishing in PLOS One.  I would, however, raise a few other issues which bothered me while reading: 1) I have not found any information on availability of the software

In addition to the note in the PLOS submission package, we have now also added a link to the end of the “Materials and methods” section.

 2) References seem a bit outdated, especially I have not found any discussion with regard to more general methods for modeling proteins. Although I understand that they have a different goal but (see below)

There has been an explosion in both the success and the utilization of machine learning modeling of protein tertiary structure in the time since our work was completed. We have added mention of a little of it; but, as you allude to, while it is tremendously exciting and transformative work, it is not directly what we are trying to do here, and we want to avoid further expanding the scope of an already large manuscript.

 3) With regard to the release of very accurate tool Alpha Fold 2, I am wondering if such general tools could not be very useful for predicting protein stability. For example, based on the deviation of modeled structure and general (secondary structure) assumptions for the modeled protein. Please discuss.

This is a compelling idea, and we have added mention of it to the discussion. One of the contributions of our manuscript is a large number of new stability scores, which could be used to validate or calibrate such an approach. One impediment to such an approach, however, is the substantial computational resources it would require.

---

## [Decision Letter · Decision Letter 1]

21 Feb 2022

Large-scale design and refinement of stable proteins using sequence-only models

PONE-D-21-36596R1

Dear Dr. Singer,

We’re pleased to inform you that your manuscript has been judged scientifically suitable for publication and will be formally accepted for publication once it meets all outstanding technical requirements.

Kind regards,

Jean-Christophe Nebel, Ph.D

Academic Editor

PLOS ONE

Additional Editor Comments (optional):

Reviewers' comments:

Reviewer's Responses to Questions

**Comments to the Author**

1. If the authors have adequately addressed your comments raised in a previous round of review and you feel that this manuscript is now acceptable for publication, you may indicate that here to bypass the “Comments to the Author” section, enter your conflict of interest statement in the “Confidential to Editor” section, and submit your "Accept" recommendation.

Reviewer #1: All comments have been addressed

Reviewer #2: All comments have been addressed

Reviewer #3: All comments have been addressed

2. Is the manuscript technically sound, and do the data support the conclusions?

Reviewer #1: Yes

Reviewer #2: Yes

Reviewer #3: Yes

3. Has the statistical analysis been performed appropriately and rigorously? 

Reviewer #1: Yes

Reviewer #2: Yes

Reviewer #3: Yes

4. Have the authors made all data underlying the findings in their manuscript fully available?

Reviewer #1: Yes

Reviewer #2: Yes

Reviewer #3: Yes

5. Is the manuscript presented in an intelligible fashion and written in standard English?

Reviewer #1: Yes

Reviewer #2: Yes

Reviewer #3: Yes

6. Review Comments to the Author

Reviewer #1: The authors have responded very well regarding my questions and those of the other reviewers. Rapid publication is recommended.

Reviewer #2: Thank you for addressing my comments and suggestions. I have no further suggestions/comments/corrections.

Reviewer #3: The authors responded to all issues raised in the review. I am satisfied with the current submission and recommend it for publication in PLOS One.

7. PLOS authors have the option to publish the peer review history of their article (what does this mean?). If published, this will include your full peer review and any attached files.

Reviewer #1: No

Reviewer #2: No

Reviewer #3: No

---

## [Editor Report · Acceptance letter]

3 Mar 2022

PONE-D-21-36596R1 

Large-scale design and refinement of stable proteins using sequence-only models 

Dear Dr. Singer:

I'm pleased to inform you that your manuscript has been deemed suitable for publication in PLOS ONE. Congratulations! Your manuscript is now with our production department. 

Kind regards, 

on behalf of

Prof. Jean-Christophe Nebel 

Academic Editor

PLOS ONE